# Structural basis for synthase activation and cellulose modification in the *E. coli* Type II Bcs secretion system

Itxaso Anso [1,2,3], Samira Zouhir[1,2,4], Thibault Géry Sana [1,2] & Petya Violinova Krasteva [1,2] ✉

Bacterial cellulosic polymers constitute a prevalent class of biofilm matrix exopolysaccharides that are synthesized by several types of bacterial cellulose secretion (Bcs) systems, which include conserved cyclic diguanylate (c-di-GMP)-dependent cellulose synthase modules together with diverse accessory subunits. In *E. coli*, the biogenesis of phosphoethanolamine (pEtN)-modified cellulose relies on the BcsRQABEFG macrocomplex, encompassing inner-membrane and cytosolic subunits, and an outer membrane porin, BcsC. Here, we use cryogenic electron microscopy to shed light on the molecular mechanisms of BcsA-dependent recruitment and stabilization of a trimeric BcsG pEtN-transferase for polymer modification, and a dimeric BcsF-dependent recruitment of an otherwise cytosolic $BcsE_2R_2Q_2$ regulatory complex. We further demonstrate that BcsE, a secondary c-di-GMP sensor, can remain dinucleotide-bound and retain the essential-for-secretion BcsRQ partners onto the synthase even in the absence of direct c-di-GMP-synthase complexation, likely lowering the threshold for c-di-GMP-dependent synthase activation. Such activation-by-proxy mechanism could allow Bcs secretion system activity even in the absence of substantial intracellular c-di-GMP increase, and is reminiscent of other widespread synthase-dependent polysaccharide secretion systems where dinucleotide sensing and/or synthase stabilization are carried out by key co-polymerase subunits.

Bacteria have evolved sophisticated nanomachines for the biogenesis of extracellular biofilm matrix components, which allow them to achieve cooperative multicellularity, increased fitness, and homeostasis[1–4]. Across the bacterial domain of life, and especially in Gram-negative pathogens and eukaryotic host-associated microbes, such as *E. coli* or *S. enterica* serovar Typhimurium, biofilm formation is typically controlled by the RNA-based second messenger c-di-GMP, which is able to elicit multiple pathway-specific physiological responses via spatially restrained intracellular signaling mechanisms[5].

Bacterial synthase-dependent exopolysaccharide secretion systems are prevalent c-di-GMP sensor-effectors, which incorporate modules for dinucleotide sensing, glycan polymerization, transmembrane export, synthase regulation, and polymer modification, and thus determine the physicochemical properties of the mature biofilms and their interactions with environment and/or eukaryotic hosts[4,6].

Bacterial cellulosic polymers are a widespread class of biofilm matrix exopolysaccharides that facilitate colonization of both biotic and abiotic environments and can have either beneficial or harmful

[1]Univ. Bordeaux, CNRS, Bordeaux INP, CBMN, UMR 5248, F-33600 Pessac, France. [2]Structural Biology of Biofilms Group, European Institute of Chemistry and Biology (IECB), 2 Rue Robert Escarpit, Pessac F-33600, France. [3]Department of Biochemistry and Molecular Biology, Faculty of Science and Technology, University of the Basque Country (UPV/EHU), Barrio Sarriena s/n, 48940 Leioa, Spain. [4]Present address: Laboratoire de Biologie et Pharmacologie Appliquée (LBPA), CNRS UMR8113, ENS Paris-Saclay, Université Paris-Saclay, Gif-sur-Yvette F-91190, France. ✉e-mail: pv.krasteva@iecb.u-bordeaux.fr

effects on human health and economy[6]. Examples of beneficial cellulose production include the secretion of crystalline cellulose by acetic acid bacteria, which finds an increasing number of biotechnological applications[6,7], or the secretion of acetylated cellulose by plant-colonizing biocontrol microorganisms[8,9]. Interestingly, while secretion of pEtN-cellulose by probiotic bacteria such as the *E. coli* Nissle 1917 strain can positively affect the intestinal epithelial barrier[10,11], the lack of cellulose secretion by the deadly enteroaggregative *E. coli* O104:H4 strain in vivo has been associated with increased virulence[3], indicating overall beneficial, antivirulence and/or anti-inflammatory properties for the polymer in the gut. In contrast, a combination of pEtN-cellulose and curli secretion by uropathogenic *E.coli* has been shown to increase adhesion to host bladder cells[12] and thus likely contributes to chronic urinary tract infections. Therefore, the mechanistic understanding of bacterial cellulose synthesis and modifications across species and across cellulose secretion systems can find a number of applications, from materials science through the selection or engineering of crop-protective biocontrol symbionts to the development of infection-specific antimicrobial compounds.

Bacterial cellulose is synthesized by dedicated cellulose synthase enzymes[1,6]. The latter's core fold of a glycosyl transferase and a transmembrane export domains (GT and TMD, respectively) is structurally conserved across kingdoms[6,13], however, bacterial BcsA orthologs typically incorporate an additional PilZ domain for c-di-GMP-dependent synthase regulation[14,15] (Supplementary Fig. 1a). Nevertheless, cellulose secretion and the actual polymer structure and modifications are determined not only by the ensemble of synthase modules, but also by a multitude of accessory subunits, which can assemble in several distinct types of Bcs secretion systems[4,6] (Supplementary Fig. 1b). In particular, type I Bcs systems are characterized by the expression of BcsD proposed to engage in a variety of intracellular scaffolds for both crystalline and modified cellulose secretion, type II systems feature the secondary c-di-GMP sensor BcsE and the pEtN-transferase BcsG components discussed below, and type III systems lack all BcsD, BcsE and BcsG subunits and often feature BcsK (periplasmic scaffolding only) rather than BcsC (scaffolding and outer membrane export) homologs in the periplasm[4,16]. In addition to the pEtN-modification conferred by BcsG in some Bcs secretion systems, others have been proposed to secrete acetylated cellulose thanks to a co-expressed alginate acetylation-like Wss complex[4,16] (Supplementary Fig. 1b).

In *E. coli* and other enterobacteria, the biofilm matrix is composed primarily of proteinaceous fimbriae, such as non-motile flagella and amyloid curli, and of cellulosic polymers produced by an *E. coli*-like or type II Bcs secretion system, which incorporates additional c-di-GMP-sensing and polymer modification modules[3,6] (Fig. 1a). In particular, *E. coli* cellulose biogenesis requires the concerted expression of two adjacent *bcs* operons (*bcsRQABZC* and *bcsEFG*), whose protein products assemble into a multicomponent BcsRQABEFG synthase macrocomplex embedded in the inner membrane, a periplasmic cellulase (BcsZ) and an outer membrane porin with periplasmic scaffolding repeats (BcsC)[17] (Fig. 1a). Whereas in vitro cellulose synthesis can be carried out with only the BcsA synthase and a C-terminal tail-anchor (TA) from the co-polymerase BcsB, micromolar concentrations of activating c-di-GMP, bivalent ions (e.g., Mg++) and uridine diphosphate glucose (UDP-glucose) as energetically preloaded substrate[18], the rest of the Bcs macrocomplex components are either essential or act as enhancers for cellulose biosynthesis in vivo[17]. Of these, the BcsG subunit has been shown to interact with a *E. coli* type-specific N-terminal domain of the BcsA synthase[17] and to introduce phosphoethanolamine (pEtN) moieties onto the nascent polymer via a pEtN-transferase domain in the periplasm[19]; the transmembrane peptide BcsF has been shown to recruit the secondary c-di-GMP-sensing protein BcsE[20]; and the latter—together with an essential-for-secretion BcsRQ ATPase complex—has been shown to form a cytosolic vestibule around the

synthase's intracellular modules[17,21] (Fig. 1a). Whereas fragmentary insights into Bcs macrocomplex formation and components' structures have been obtained from several crystallographic and electron microscopy studies[17,20–22], the overall stoichiometry, assembly and regulatory mechanisms have remained enigmatic.

Here we use cryogenic electron microscopy (cryo-EM) to settle conflicting reports on the macrocomplex stoichiometry[21,22] and reveal the molecular mechanisms of regulatory subunit recruitment and function. We demonstrate that BcsA's N-terminal domain adopts an amphipathic fold to recruit three copies of the pEtN-transferase BcsG, stabilized opposite of the previously reported hexameric BcsB crown. We further demonstrate that the single-pass inner membrane polypeptide BcsF folds into an X-shaped dimer to recruit and retain an asymmetric $BcsE_2R_2Q_2$ complex around the synthase's cytosolic modules. In this so-formed vestibule, the N-terminal domain of a BcsR protomer plugs into a hydrophobic pocket at the $BcsA^{GT-PilZ}$ domain interface, and BcsQ buttresses the PilZ domain, likely stabilizing the catalytically competent state. Finally, we demonstrate that through a composite, interdomain c-di-GMP binding site BcsE acts as a higher-affinity dinucleotide sensor that can adopt discrete dimerization interfaces to maintain the activating vestibule components even in the absence of direct c-di-GMP-BcsA interactions. Together, our structural data suggest that the *E. coli*-like Type II Bcs secretion systems have evolved a cooperative activation-by-proxy mechanism to lower the threshold for c-di-GMP-dependent activation, as well as an additional synthase module for pEtN-transferase recruitment and efficient co-synthetic polymer modification.

## Results

### Stoichiometry of the assembled Bcs macrocomplex of *E. coli*

Bacterial BcsA orthologs are processive GT2 family synthases with a single cytosolic GT domain that uses UDP-glucose as substrate in an inverting, divalent metal ion-dependent mechanism of glycan polymerization, best studied in vitro in the *Rhodobacter sphaeroides* BcsAB heterodimeric complex[15,23,24]. Polymerization is coupled with inner membrane polysaccharide extrusion through a narrow pore in the $BcsA^{TMD}$-$BcsB^{TA}$ inner membrane complex translocating a non-modified, non-hydrated homopolymer. In the resting state, the $BcsA^{GT}$ active site is capped by a so-called gating loop, whose conformation is stabilized by interactions with the N-proximal $BcsA^{PilZ}$ domain linker that senses c-di-GMP[23]. In the presence of micromolar concentrations of dinucleotide the PilZ domain undergoes an -18° rotation and 4.4 Å displacement around a C-proximal α-helical hinge, and the linker-gating loop-stabilizing interactions are released to yield a catalytically competent state[15]. Processive cycles of active site opening, substrate entry, gating loop closure, polymerization, and translocation are then determined by the presence of product vs. substrate in the active site and minute movements of a so-called finger helix in the bottom of the enzyme's active site[6,24].

Whereas BcsA itself is highly conserved, the secretion-competent synthase macrocomplexes are strikingly diverse across the bacterial clade[4,6]. In *E. coli*, in particular, the catalytic BcsAB tandem has been shown to associate with the ensemble of the inner membrane and cytosolic subunits in an approximately megadalton-sized secretory assembly. In it, the synthase associates in a non-canonical BcsA:BcsB stoichiometry with up to six BcsB copies whose donut-shaped periplasmic modules assemble into a superhelical crown with stacking carbohydrate-binding modules likely guiding the extruded polysaccharide into the periplasm and towards the BcsC periplasmic scaffold[17,21]. Additionally, the synthase has been found to associate stably with an essential-for-secretion BcsRQ tandem, with the secondary c-di-GMP-sensing protein BcsE, the inner membrane polypeptide BcsF and the pEtN-transferase BcsG[17,21,22] (Fig. 1a). The stoichiometry and recruitment mechanisms of all of these latter components have remained under debate, mostly due to

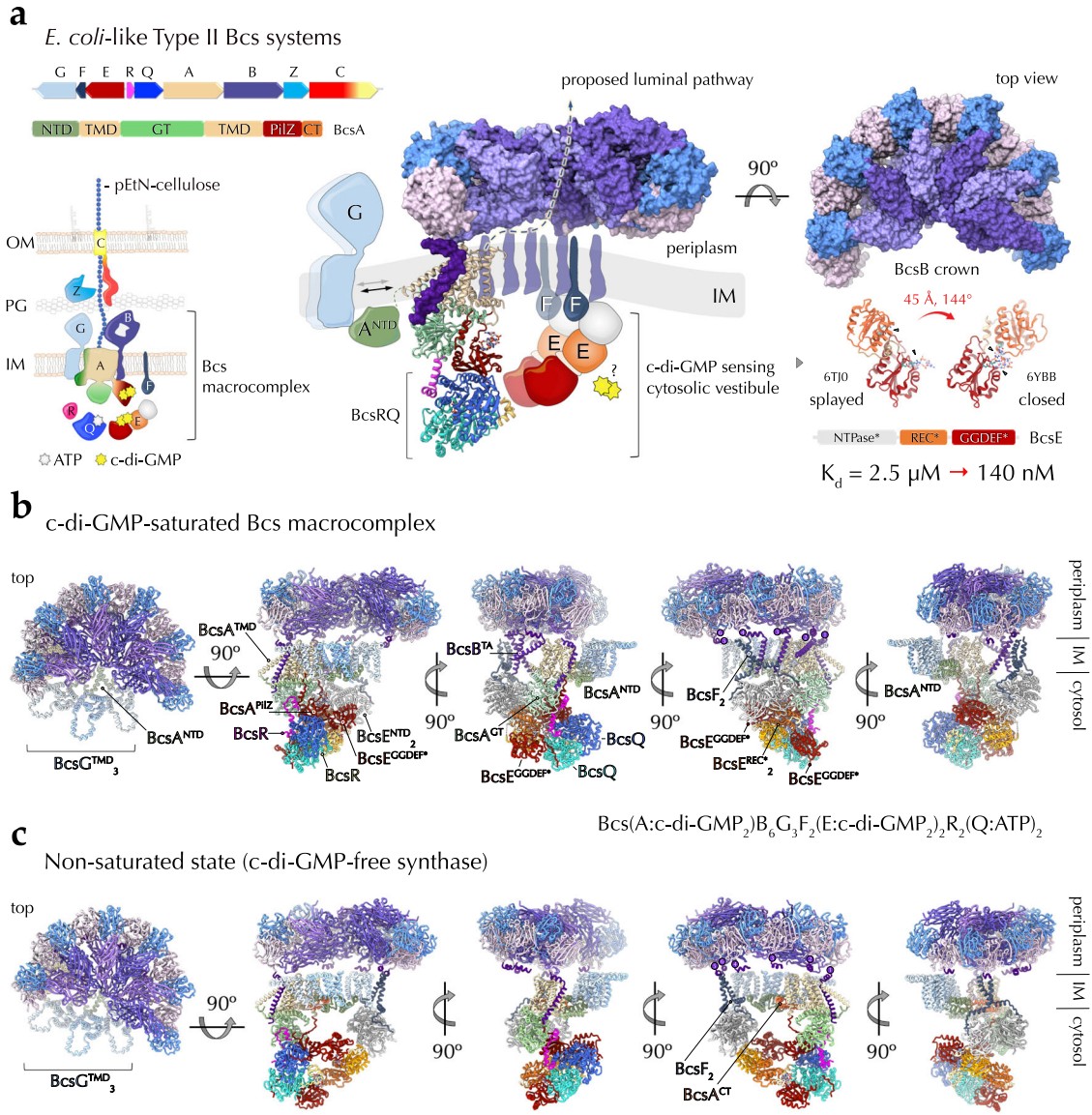

**Fig. 1 | State-of-the-art and here-in presented structures of the Bcs macro-complex from the *E. coli* Type II cellulose secretion system. a** Left, *E. coli bcs* operon organization, BcsA domain architecture and thumbnail representation of the secretion system topology in the *E. coli* envelope. Middle and right, current structural insights into complex assembly from X-ray crystallographic and electron microscopy structures[17,20–22]. Adapted with modifications from Krasteva 2024[4] under the CC BY 4.0 license (https://creativecommons.org/licenses/by/4.0/legalcode). NTD N-terminal domain (green), TMD transmembrane domain (wheat), GT glycosyl transferase domain (light green), PilZ c-di-GMP-sensing PilZ domain (dark red), CT C-terminal tail with amphipathic helices (orange), OM outer membrane, PG peptidoglycan, IM inner membrane, ATP adenosine triphosphate, c-di-GMP cyclic diguanylate, pEtN phosphoethanolamine, NTPase* (light gray) catalytically incompetent nucleoside triphosphatase domain, REC* (orange) phosphorylation-incompetent receiver domain, GGDEF* (dark red) degenerate diguanylate cyclase domain. Multidomain BcsB hexamerizes to form a periplasmic crown shown in two different views. The carbohydrate-binding domains are shown in shades of light purple, the flavodoxin-like domains in blue and pink, and the C-terminal tail-anchor (TA) in dark purple. Densities for BcsA[NTD] (green), BcsG (light blue), BcsE (tricolor) and BcsF (dark blue) have remained practically unresolved in the macrocomplex and are represented as thumbnails, whereas crystallographic snapshots have captured two different conformations of BcsE, shown on the right[20,21]. The relative REC* domain displacement and rotation are indicated (45 Å and 144 degrees, respectively). The formation of a composite c-di-GMP binding site by RxxD (arginine-two residues-aspartate) motifs from both the degenerate REC* and GGDEF* domains increases the affinity for dimeric c-di-GMP from the low micromolar to nanomolar range[21] (bottom right). **b** Cartoon representations of the here-in-resolved cryo-EM structure of the assembled, c-di-GMP-saturated Bcs macrocomplex in five different views. **c** Cartoon representations of the here-in-resolved cryo-EM structure of the assembled Bcs macrocomplex featuring a c-di-GMP-free BcsA.

the limited resolutions of previously reported structural models in the literature[17,21,22].

Here we present cryo-EM structures of the *E. coli* Bcs macro-complex positioning all seven BcsRQABEFG partners and multiple c-di-GMP-binding sites (Fig. 1b, c and Supplementary Fig. 2). We show that the Bcs macrocomplex contains a single BcsA synthase, which associates in the membrane with hexameric BcsB on one side and a trimeric

BcsG pEtN-transferase complex on the other. We further show that BcsA's catalytic and c-di-GMP-sensing domains engage in extensive cytosolic interactions with the essential-for-secretion BcsRQ complex, present as a heterotetrameric BcsR$_2$Q$_2$ assembly. The latter is further retained by direct interactions between BcsQ and the C-terminal modules of dimeric BcsE, whose degenerate receiver (REC*) and diguanylate cyclase (GGDEF*) domains bind an intercalated dimeric c-

di-GMP per BcsE protomer. The previously uncharacterized N-terminal domains (NTD) of BcsE, on the other hand, form a membrane-proximal head-to-head dimer of P-loop nucleotide triphosphatase (NTPase)-like modules. Remarkably, the latter's central β-sheets are complemented at each distal side by an additional β-strand from the extended cytosolic tails of an inner membrane-embedded BcsF dimer, whose X-shaped transmembrane helices positioned near the tail-anchors of synthase-distal BcsB copies from the crown. Together, these results demonstrate a definitive $BcsR_2Q_2AB_6E_2F_2G_3$ stoichiometry for the assembled *E. coli* Bcs macrocomplex, which binds up to six c-di-GMP molecules for maximal synthase activation and nascent polysaccharide modification (Fig. 1b, c). Finally, we visualize the complex's intrinsic conformational plasticity, in which the regulatory BcsRQEF vestibule, and BcsE in particular, can resort to alternative protein interaction interfaces to maintain the activating BcsRQ partners onto the synthase's cytosolic modules even in non-saturating dinucleotide concentrations (Fig. 1b, c).

## The inner membrane $BcsAB_6G_3F_2$ complex

We demonstrated previously that contrary to the canonical 1:1 BcsAB assemblies observed in purified samples from *R. sphaeroides* (Type III Bcs secretion system)[23] and *G. hansenii* (Type I Bcs secretion system)[25], in the assembled *E. coli* Bcs macrocomplex BcsA associates with a BcsB hexamer whose periplasmic modules of alternating carbohydrate-binding and flavodoxin-like domains ($CBD_1$-$FD_1$-$CBD_2$-$FD_2$) polymerize via a β-sheet complementation mechanism between the $FD_1^n$:$FD_2^{n+1}$ domains of adjacent BcsB protomers[21]. The structure of the hexameric periplasmic crown is refined here to 2.35 Å resolution and reveals to near-atomic detail the molecular mechanism of BcsB polymerization, with more than 3300 Å$^2$ interface surface and a free energy gain of −22.5 kcal/mol between each pair of adjacent protomers (Supplementary Fig. 3).

Whereas earlier studies have visualized the overall fold of the transmembrane regions of the BcsA synthase and the C-terminal tail-anchor of the co-polymerase BcsB[1] protomer[21,22], here we position the transmembrane anchors for most of the remaining BcsB subunits and present the first structures of the regulatory BcsG and BcsF partners at side-chain resolution. The first and second BcsB copies engage in contacts with the BcsA synthase, where the C-terminal TA of BcsB[1] fits in a groove formed by BcsA TMα1−3 and is further encased by BcsA$^{NTD}$ to complete the membrane export module, while BcsB$^2$-TA engages in limited hydrophobic contacts with the loop connecting BcsA TMα3 and TMα4 at the periplasmic side of the inner membrane. Near the synthase-distal BcsB copies, on the other hand, positions an X-shaped transmembrane BcsF dimer. In particular, the BcsF tandem positions near the fourth and third BcsB protomers in the c-di-GMP-saturated macrocomplex (Fig. 2b), and near the fifth and the fourth BcsB copies in the context of a dinucleotide-free synthase (Fig. 2c). Each BcsF subunit features a single transmembrane helix, which upon exit from the inner membrane kinks into an amphipathic helical extension and a cytosolic C-terminal tail engaged in interactions with a BcsE N-terminal domain as further described below. Interestingly, neither BcsE, nor BcsF engage in direct protein-protein contacts with their intraoperon partner BcsG.

Using bacterial two-hybrid functional complementation (BACTH) assays, we demonstrated previously that the *E. coli* type-specific N-terminal domain of the synthase interacts specifically with the BcsG enzyme in cellulo[17]. In addition, BcsG has been shown to directly affect BcsA integrity in the membrane[22,26] and in some strains to be essential for cellulose secretion[17,26]. Indeed, using local refinement to resolve the more dynamically associated pEtN-transferase (Fig. 2 and Supplementary Fig. 4), we show here that BcsA$^{NTD}$ adopts an amphipathic fold and recruits three copies of the BcsG pEtN-transferase whose transmembrane N-terminal domains are tightly packed between BcsA$^{TMD}$ and the sixth BcsB protomer of the crown, whereas the C-terminal

catalytic BcsG modules remain unresolved in the structures. The BcsA N-terminus folds into a W-shaped series of amphipathic α−helices whose connecting loops coordinate the BcsG protomers via conserved amino acid motifs in an otherwise weakly conserved primary structure (Fig. 2a−c). Each of the BcsG$^{NTD}$ folds into 5 transmembrane helices (TMα1−5, which anchor the protein in the inner membrane and, via the TMα4−TMα5 connecting loop interact with BcsA$^{NTD}$, which in turn packs against an α-helical amphipathic hairpin formed by the BcsA C-terminus (Fig. 2a−c). The BcsG TMα3−TMα4 linker region, on the other hand, folds into a short amphipathic α-helical loop at the periplasmic membrane interface, whereas TMα5 is predicted to extend into a 48 residue-long flexible linker[27], followed by a crystallographically characterized[26,28] but here unresolved C-terminal pEtN-transferase domain (Fig. 2a, d). Based on homology with other alkaline phosphatases, such as the *Neisseria meningitidis* lipid A pEtN-transferase EptA (*Nm*EptA)[29], the amphipathic helical loop and the extended interdomain linker could potentially assist the C-terminal catalytic domain in substrate-extraction by interactions with the polar headgroups of periplasm-facing phospholipids (phosphatidyl-ethanolamine (PE) in the case of BcsG) and/or could allow for significant conformational flexibility in substrate delivery to the target acceptor. Interestingly, in BcsG the amphipathic helical loops point outwards relative to the crown's lumen, where BcsB's stacked carbohydrate-binding domains are proposed to form the polysaccharide extrusion path (Fig. 1). This suggests major conformational gymnastics of the catalytic C-terminal domains for pEtN extraction and transfer onto the nascent cellulosic polymer (Fig. 2e), and could potentially explain the lack of resolved BcsG$^{CTD}$-corresponding regions in the averaged electron density maps.

Remarkably, the presence of three BcsG copies is in contrast with a previous assignment of densities from a low-resolution cryo-EM map of the macrocomplex to a dimeric BcsG enzyme[22] and most of the reported mechanistic studies on active pEtN-transferases (including on the C-terminal periplasmic module of BcsG) present no substrate- or product-determined prerequisite for catalytic domain oligomerization[26,28−34]. This suggests that the three BcsG copies visualized here likely act independently from each other to dynamically sample the membrane for, extract, and transfer pEtN moieties from inner membrane PE onto the nascent polysaccharide (Fig. 2e). Importantly, while this work was under review a separate study reported independently the recruitment of trimeric BcsG via BcsA$^{NTD}$, based on lower-resolution cryo-EM data, subcomplex purification and AlphaFold modeling[35]. Together, these results further validate the experimental structural data presented here, and the two studies integrate and redress the structure-function model of pEtN-transferase association and function.

## The BcsF:BcsE interactions for cytosolic complex recruitment

We previously demonstrated that the cellulose secretion enhancer BcsE can form equimolar $BcsE_2R_2Q_2$ complexes with the essential-for-secretion BcsRQ tandem in solution, that BcsE is sequestered by BcsF to the membrane and that BcsE's N-terminal domain is necessary for stable cytosolic complex association with the synthase macrocomplex[20]. Nevertheless, how BcsE and BcsF interact, what structures they adopt in the secretory assembly, and even their actual membrane-bound stoichiometries have remained unresolved[21,22].

Here we show BcsE and BcsF interact in an asymmetric and heterotetrameric $BcsE_2F_2$ complex (Fig. 1b, c and Fig. 3a). In particular, BcsF adopts an X-shaped dimeric conformation within the inner membrane, stabilized by a hydrophobic N-proximal transmembrane interface burying 626 Å$^2$ of surface area with free energy gain of −15.5 kcal/mol (Fig. 3b). At the C-termini, each BcsF protomer recruits a BcsE partner copy via cytosolic β-sheet complementation interactions with the central 9-stranded β-sheet of the interacting BcsE$^{NTD}$ (Fig. 3b). The BcsF C-terminal tail threads along a shallow hydrophobic patch

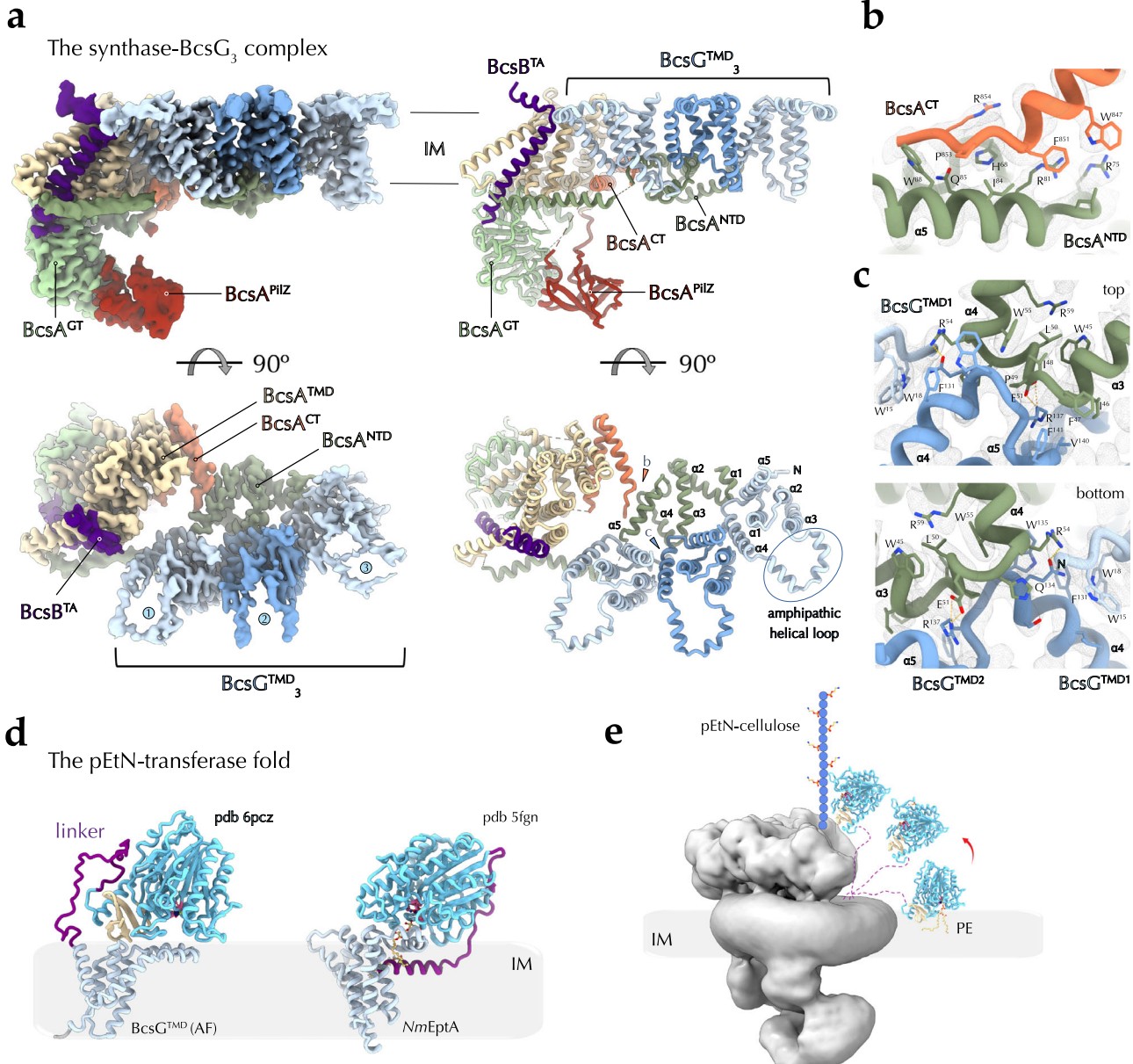

**Fig. 2 | Cryo-EM structure of the synthase:pEtN-transferase complex. a** Different views of a locally refined cryo-EM structure of the c-di-GMP-free BcsA-BcsB$^{TA}$-BcsG$_3$ assembly (BcsAG$_3$ for simplicity) with corresponding electron densities (left) and cartoon representation (right). **b, c** Zoom-ins on the specific protein-protein interfaces with key residues shown as sticks and the electron density as a mesh. **d** Composite predicted structure of full-length BcsG (catalytic domain: X-ray structure of the *E. coli* BcsG$^{CTD}$; NTD and linker, AlphaFold (AF)) and crystal structure of the lipid A pEtN-transferase from *Neisseria meningitidis* EptA. The flexible interdomain linkers are colored in purple. **e** Model for independent function of the three BcsG copies for substrate-extraction and cellulose modification. IM inner membrane, PE phosphatidyl-ethanolamine.

onto BcsE's degenerate NTPase* domain and provides an overall charged solvent-exposed surface for the assembly (~837 Å buried interface with free energy gain of −12.9 kcal/mol) (Fig. 3c). Consistent with the observed complex, BcsF truncations before or after P$^{43}$ preceding the C-terminal cytosolic tail lead to incomplete Bcs macro-complex assembly and corroborate the requirement for stable BcsF-BcsE$^{NTD}$ interaction for vestibule complex recruitment (Fig. 3c and Supplementary Fig. 5). Importantly, the observed β-sheet complementation mechanism for BcsF-driven BcsE recruitment and BcsE$^{NTD}$ dimerization (see below) is likely conserved across enterobacteria as shown in ColabFold and AlphaFold3-predicted models of a consensus BcsE$_2$F$_2$ complex derived from representative homologs across the enterobacterial clade (Supplementary Fig. 6).

The cryo-EM structures presented here are consistent with the previously characterized tripartite architecture of BcsE, comprising a degenerate trio of an NTPase*, REC*, and GGDEF* domains (Fig. 1a). Nevertheless, rather than engaging in head-to-tail interactions as proposed previously based on indirect BACTH interaction assays[20,21], the two BcsE$^{NTD}$ modules pack against each other in a head-to-head dimer, stabilized primarily by hydrophobic and π−stacking interactions in the center and by the peripheral BcsF C-terminal tails at the periphery (747 Å buried with free energy gain of −2.7 kcal/mol at the BcsE$^{NTD}$ dimer interface) (Fig. 3d).

The REC*-GGDEF* domain tandem interacts with BcsQ via an extended C-terminal tail trailing along the BcsQ surface, as observed in crystallographic snapshots previously[21]. However the REC* domains,

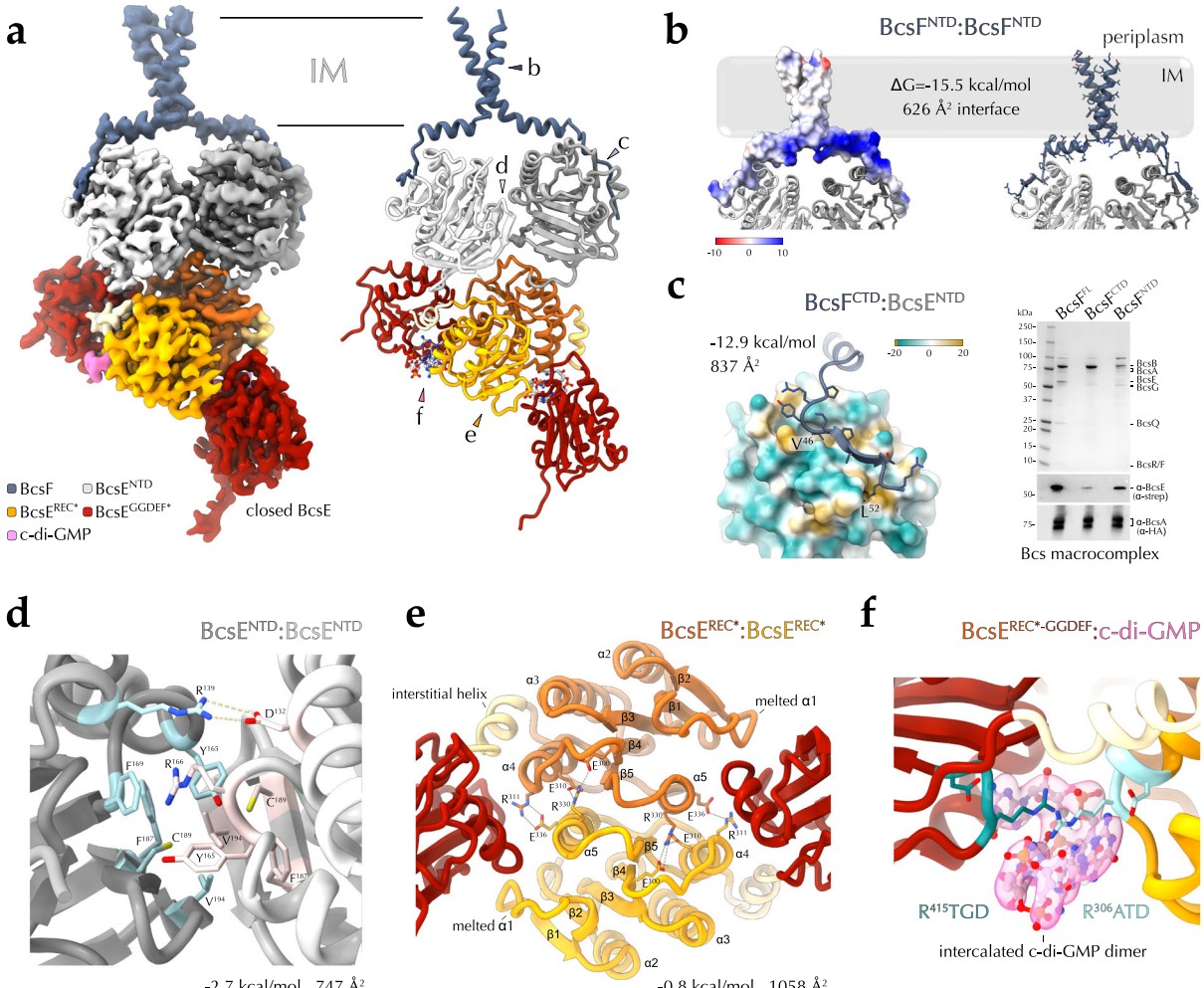

**Fig. 3 | BcsF-dependent BcsE recruitment and regulatory complex conformation in the c-di-GMP-saturated state. a** Locally refined cryo-EM structure of the BcsE₂F₂ assembly from the c-di-GMP-saturated synthase macrocomplex shown as electron density and in cartoon. IM, inner membrane. **b** BcsF dimerization shown as Coulombic electrostatic potential-colored surface (left, default −10 to 10 range) and in cartoon and sticks (right). **c** BcsE-BcsF interactions. Left, BcsE^NTD is shown as a lipophilicity-colored surface (default −20 to 20 range), BcsF residues−including the hydrophobic plug residues V⁴⁶ and L⁵²−are shown as sticks. Right, recombinant expression and purification of the Bcs macrocomplex with various BcsF variants (Bcs^HisRQA^HA-FLAGB + Bcs^strepEF*G). Protein-specific bands are identified as previously[17,21]. BcsE and BcsA-specific signals are further detected by western blotting with epitope tag-specific antibodies in the bottom (representative data from three independent experiments). **d** The BcsE^NTD dimerization interface. **e** The BcsE^REC* dimerization interface. **f** The c-di-GMP-binding dual I-site pocket in closed BcsE. All interface parameters were calculated with PISA[57].

which are not in contact in the crystallized states, engage in head-to-head dimerization interactions mediated by a α4−β5−α5 interface (Fig. 3e), observed as a canonical REC domain dimerization interface in many phosphorylation competent response regulators[36,37]. An intercalated c-di-GMP dimer is found at each *cis*-interdomain interface of a closed BcsE[21], stabilized by a composite R³⁰⁶ATD-R⁴¹⁵TGD I-site tandem contributed by the corresponding REC* and GGDEF* domains, respectively (see below) (Fig. 1a and Fig. 3f). Finally, the two GGDEF* domains adopt different orientations relative to the apical BcsR₂Q₂ tandem consistent with the overall macrocomplex asymmetry. In the c-di-GMP-saturated macrocomplex, one BcsE^GGDEF* copy adopts an overall interaction interface consistent with the previously reported crystallized states and contacts BcsA^GT via its REC* module. The second BcsE^GGDEF*, on the other hand, positions above the BcsQ dimer interface and is further stabilized by the β-strand connecting loops at the bottom of the BcsA^PilZ domain barrel (Fig. 3a). In the macrocomplex featuring a c-di-GMP-free synthase, the relative orientation of the REC* and GGDEF* BcsE modules are yet different and discussed in detail below.

## The activating synthase:BcsRQ interactions

We previously showed that, upon co-expression, BcsR and BcsQ stabilize and act as chaperones to each other via the formation of a heterotetrameric BcsR₂Q₂ complex[20] with essential roles in Bcs system positioning, assembly, stability, and function[17,21,38]. Using X-ray crystallography and cryo-EM, we positioned the latter at the apical densities of the cytosolic vestibule formed around the synthase's PilZ domain, however, the limited electron density map resolution prevented us from deciphering the specific protein-protein interactions and their roles in cellulose secretion[21]. Here we locally refined the structure of the crownless Bcs macrocomplex complex to an average resolution of 2.85 Å, visualizing all interaction interfaces and coordinated nucleotide co-factors. An assymetric BcsR₂Q₂ complex is recruited to the membrane complex via BcsE's C-terminal elongated tails, where both BcsQ copies interact with the synthase's PilZ module (Fig. 4a, b) and adopt the nucleotide-driven sandwich dimer conformation characteristic for the SIMIBI (SIgnal recognition particle, MinD, and BioD) family of protein-sorting NTPases to which BcsQ belongs[21]. Consistent with the previously reported crystal structures of

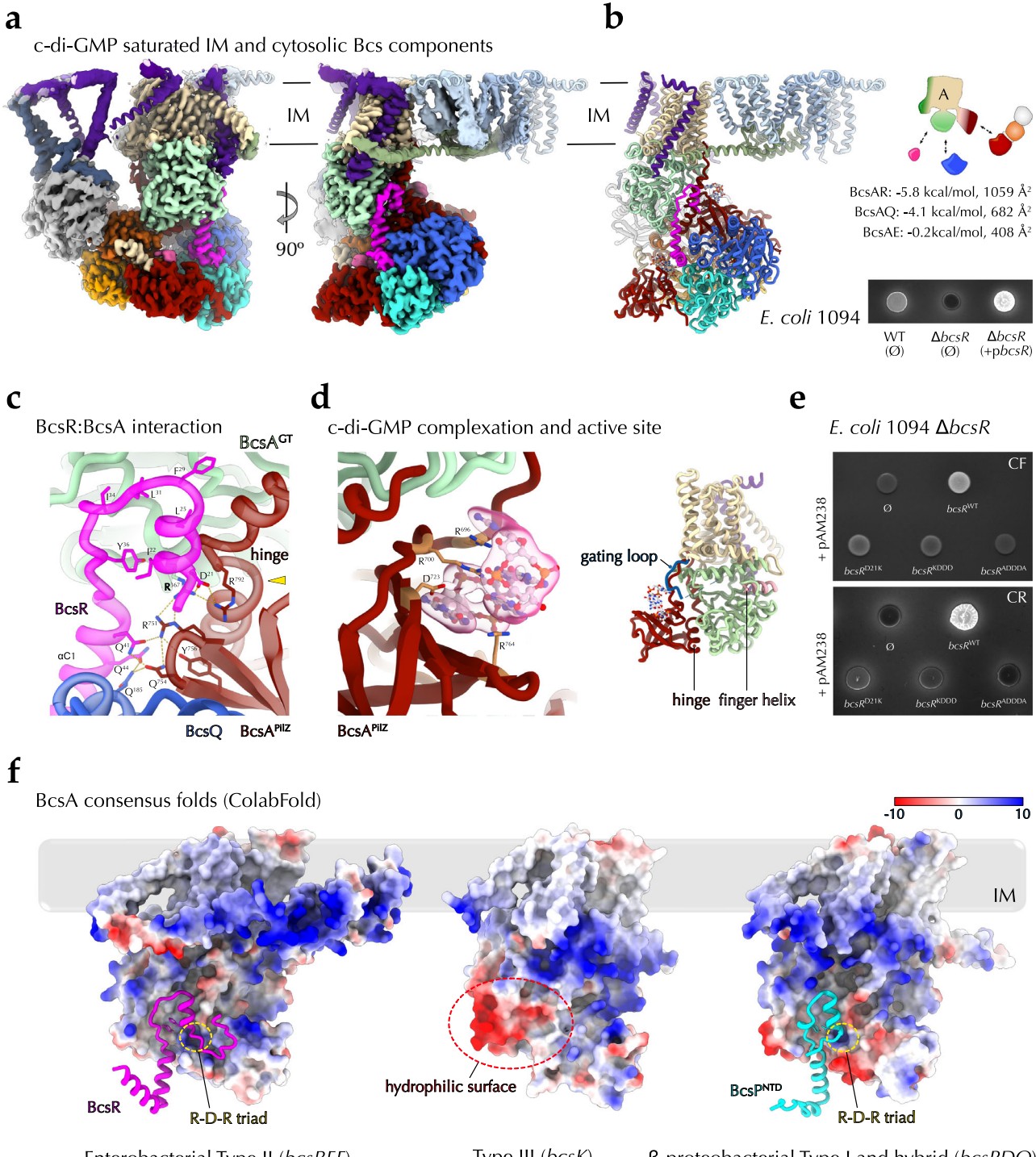

**Fig. 4 | The c-di-GMP-bound synthase macrocomplex. a** Locally refined cryo-EM map and fitted structure of the crownless c-di-GMP-saturated synthase macrocomplex in two different views. IM inner membrane. **b** Cartoon representation of the same assembly, summary of the BcsA interactions with the cytosolic vestibule partners and stimulatory effects of BcsR overexpression as detected by binding and UV-fluorescence of *E. coli* macrocolonies grown on Congo Red-supplemented plates. **c** A zoom-in on the BcsA-BcsR interface with key residues shown as sticks. The R-D-R triad is indicated with a yellow arrowhead. **d** c-di-GMP coordination, together with its corresponding electron density, and overall core synthase fold showing unstructured gating loop and an accessible active site. **e** Effects on cellulose secretion upon BcsR$^{NTD}$ mutagenesis using plasmid-based complementation

with various BcsR mutants. KDDA D$^{21}$K-L$^{25}$D-F$^{29}$D-L$^{31}$D, ADDDA D$^{21}$A-L$^{25}$D-F$^{29}$D-L$^{31}$D-Y$^{36}$A. CR Congo Red, CF calcofluor. Data representative of three independent experiments with two biological replicates each. **f** Consensus Colab-Fold structural models of Type II BcsA-BcsR (based on multiple BcsA homologs encoded by *bcsR*- and *bcsEF*-positive enterobacterial *bcs* clusters), Type III BcsA (derived from *bcsK*-positive *bcs* clusters) and Type I and hybrid BcsA-BcsP$^{NTD}$ (derived from *bcsPDQ*-positive *bcs* clusters). BcsA is shown as Coulombic electrostatic potential-colored surface, and BcsR (magenta) and BcsP$^{NTD}$ (cyan) are shown in cartoon. The stabilizing pairs of hydrophobic residues in BcsR and BcsP$^{NTD}$ are shown as sticks.

the BcsR$_2$Q$_2$ complex[21], the BcsR copies stabilize the ATP-bound BcsQ apical dimer via their V-shaped C-terminal tandem of α-helices (αC$_1$ and αC$_2$). Importantly, whereas one of the BcsR protomers is solvent-exposed and features an unresolved N-terminal domain, the other BcsR copy also interacts with the back of the BcsA catalytic module (Fig. 4a–c). Contrary to the crystallized states where BcsR$^{NTD}$ can adopt a β-hairpin conformation that threads onto the surface of dimeric BcsQ (Supplementary Fig. 7), here residues D$^{21}$-S$^{30}$ fold into an N-proximal α-helix (αN) that U-turns into an extended linker before adopting the V-shaped C-terminal domain onto the BcsQ dimer interface (Fig. 4c and Supplementary Fig. 7). The resulting N-terminal hairpin nestles into a hydrophobic BcsA$^{GT}$ pocket via a L$^{25}$-F$^{29}$-L$^{31}$-I$^{34}$ plug at the tip and a I$^{22}$:Y$^{36}$ stabilizing interaction at the base. The latter isoleucine:tyrosine pair are thus brought to a distance of less than 4 Å, as compared to the ~40 Å that separates them in the BcsQ-interacting crystallized state (Supplementary Fig. 7). The strictly conserved D$^{21}$ positions between R$^{367}$ from the BcsA$^{GT}$ domain and R$^{792}$ in the middle of the C-proximal hinge that enables PilZ rotation upon BcsA:c-di-GMP complexation (Fig. 4c). Overall, the BcsR:BcsA interaction interface buries 1059 Å and contributes a free energy gain of −5.8 kcal/mol (Fig. 4b, c). The BcsA$^{PilZ}$ domain orientation is further stabilized by interactions between the β4–β5 connecting loop of the PilZ barrel and N-proximal residues from BcsR-αC$_1$, as well as by an extensive interface with the underlying BcsQ protomer (682 Å buried with a free energy gain of −4.1 kcal/mol) (Fig. 4b, c). On the other side of the β-barrel, an intercalated c-di-GMP dimer is found coordinated between the arginines from the canonical R$^{696}$RxxR motif in the N-proximal PilZ domain linker, the active site gating loop is unstructured, and the active site is substrate-accessible (Fig. 4d). Together, the BcsEF-stabilized BcsRQA complex appears to induce or stabilize the synthase into a catalytically competent state, which is consistent with previous in vitro activity data demonstrating dramatic synthase activation in the presence of excess cytosolic vestibule components, with stimulatory effects observed even in the absence of c-di-GMP[22]. Consistent with the observed BcsR:BcsA interactions, plasmid-based overexpression of BcsR leads to overproduction of matrix pEtN-cellulose (Fig. 4b), whereas mutations in the N-terminal domain, which do not affect BcsRQ complex formation per se[21], led to severe or complete loss of pEtN-cellulose secretion (Fig. 4e).

The experimentally determined BcsR:BcsA interaction interface via a surface-exposed hydrophobic pocket at the back of the synthase's GT and PilZ modules is likely conserved across the enterobacterial clade. Indeed, a ColabFold model of a consensus BcsRA complex based on protein sequences from representative cellulose-secreting enterobacteria demonstrates an overall conserved BcsR fold and interface residues, including a hydrophobic plug at the tip, stabilizing F:Y π–stacking interactions at the base of the BcsR$^{NTD}$ hairpin (corresponding to the isoleucine:tyrosine pair discussed above), and a conserved R:D:R triad at the GT:BcsR:hinge interface (Fig. 4f). These are accompanied by a hydrophobic BcsR-binding surface pocket on BcsA, supporting synthase-partner activator coevolution. Interestingly, similar fold prediction based on a consensus BcsA sequence derived from homologs encoded by bcsK-containing Type III bcs clusters which typically lack cytosolic Bcs regulators[4,16] lacks a corresponding hydrophobic pocket despite overall high conservation of the BcsA sequence and fold (Fig. 4f).

We recently showed that most β-Proteobacteria featuring bcsD in a Type I or hybrid bcs operon architecture, also encode proline-rich BcsP homologs[39] (Supplementary Fig. 1b). Similarly to the proline-rich cellulose crystallinity factor BcsH/CcpAx from Gluconacetobacter hansenii, which determines the formation of a longitudinal BcsHD cytoskeletal scaffold (a.k.a. cortical belt) and the respective linear alignment of the synthase terminal complexes for cellulose secretion and crystalline ribbon formation[40,41], β-proteobacterial proline-rich BcsP recruits BcsD into distinct cytoskeletal assemblies that are key to

cellulose biogenesis and the mature biofilm architecture[39]. Interestingly, the N-terminal regions of BcsP homologs show homology to enterobacterial BcsR[16] and, similarly to the latter, BcsP expression and/or stability appeared enhanced in the presence of co-expressed and interacting BcsQ[39]. We, therefore, retrieved multiple sequences of representative and co-occurring β-proteobacterial BcsA and BcsP homologs and modeled the consensus complex between the synthase and BcsP$^{NTD}$. Indeed, the predicted structure confirms both the presence of a conserved hydrophobic pocket on the synthase and a BcsR-like hairpin-shaped plug for BcsP$^{NTD}$, suggesting a common mechanism for synthase regulation among widespread Type I and Type II Bcs secretion systems (Fig. 4f).

## BcsA activation-by-proxy in non-saturating c-di-GMP

BcsE was originally defined as a GIL-, or GGDEF I-site like-, domain protein due to a conserved C-terminal region sensing c-di-GMP via an RxxD (R$^{415}$TGD in E. coli BcsE) motif similar to the product-sensing I-sites, which are found on many catalytically active diguanylate cyclases and are involved in feedback inhibition or dinucleotide signal relay[37,42]. We demonstrated previously that the so-called GIL domain is, in fact, a degenerate and conformationally dynamic REC*-GGDEF* domain tandem, where the R$^{415}$TGD sequence corresponds to the canonical I-site in an otherwise catalytically incompetent diguanylate cyclase module[20]. Whereas this motif is absolutely necessary for dinucleotide complexation, the phosphorylation-incompetent REC* domain can undergo significant conformational rearrangements to contribute a second I-site motif (R$^{306}$ATD) for an intercalated c-di-GMP dimer complexation[21] (Fig. 1a). This corresponds to a relatively compact or closed BcsE conformation observed in a BcsRQE$^{REC*-GGDEF*}$ crystal structure reported previously[21] and is also consistent with the cryo-EM structure presented above. The dissociation constants for dimeric c-di-GMP complexation thus change from the low micromolar (~2.5 μM, for the contribution of the GGDEF* I-site alone) to the nanomolar range (~140 nM, for dual I-site coordination)[21] (Fig. 1a). The latter c-di-GMP-binding affinity is significantly higher than the affinity for activating c-di-GMP complexation by the BcsA synthase itself, previously reported in the low micromolar range and orders of magnitude higher than the global cytosolic c-di-GMP concentrations in the early stages of biofilm formation[3,18,43].

This raises the question of whether and how c-di-GMP binding to the higher-affinity sensor BcsE could have stimulatory effects on synthase activity and cellulose biogenesis in non-saturating dinucleotide concentrations. One possible mechanism is that the molecular breathing of the Bcs macrocomplex during the processive cycles of glucose polymerization could cause reiterative conformational changes in BcsE and the synthase, thus leading to diametric changes in their respective dinucleotide binding affinities and c-di-GMP recycling for reiterative synthase activation. Alternatively, the higher-affinity c-di-GMP binding to BcsE, associated with the latter's compact conformation within the multicomponent cytosolic vestibule could stabilize the synthase in a catalytically competent conformation regardless of its direct dinucleotide complexation.

To gain mechanistic insights into the c-di-GMP-dependent regulation, we kept low micromolar concentrations of the dinucleotide (2–4 μM) throughout the purification procedure and prepared the cryogrids after a final fast concentration step to ~2:1 c-di-GMP:Bcs macrocomplex ratio. As these conditions are close to the predetermined dissociation constants for dimeric c-di-GMP complexation to both the BcsA and the BcsE GGDEF* domain alone (i.e., consistent with splayed BcsE without contributions of the secondary REC* domain I-site to dinucleotide binding)[18,21] but more than an order of magnitude higher than that for compact, tandem I-site-contributing BcsE[21], we hypothesized that they could allow us to capture either a BcsE-saturated/BcsA non-saturated state or, inversely, a splayed, non-saturated BcsE accompanying a c-di-GMP-bound synthase.

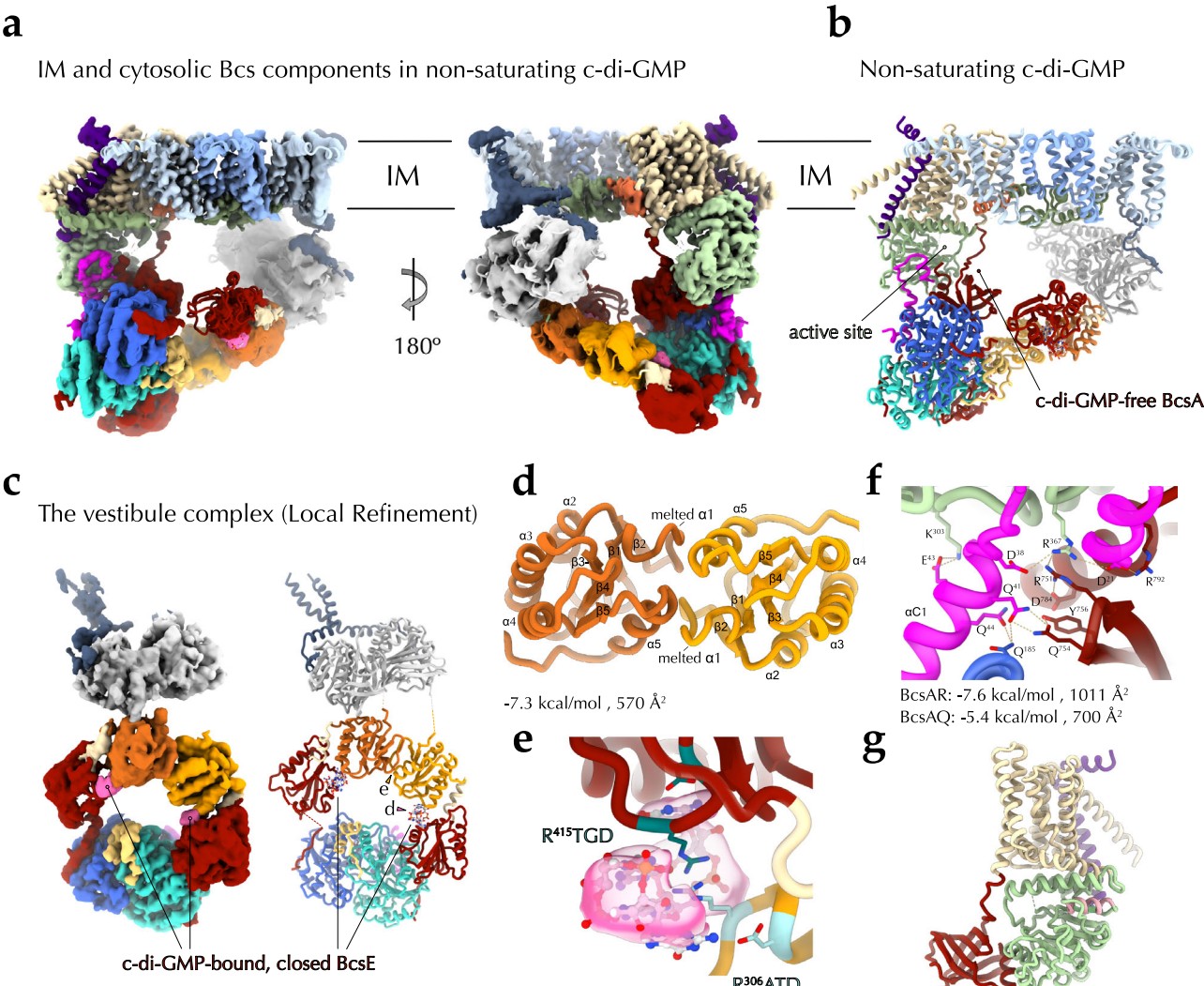

**Fig. 5 | The synthase macrocomplex in limiting c-di-GMP. a** Locally refined cryo-EM map and fitted structure of the crownless synthase macrocomplex featuring a c-di-GMP-free synthase in two different views. IM inner membrane. **b** Cartoon representation of the same assembly. **c** The cryo-EM map and model of a locally refined BcsRQEF assembly. **d** A zoom-in on c-di-GMP binding by a composite, dual I-site pocket in closed BcsE. **e** REC* domain dimerization interface in the non-saturated macrocomplex. **f** A zoom-in on the BcsA:BcsR interface and summary of the synthase's interactions with the cytosolic vestibule partners[57]. **g** Overall core synthase fold showing unstructured gating loop and an accessible active site.

About half of the structurally resolved particles featured the fully c-di-GMP-saturated state shown above, where all three c-di-GMP-sensing subunits (BcsA and BcsE$_2$) are bound to an intercalated dinucleotide dimer in a preserved 2:1 dinucleotide-to-protein binding site ratio. Interestingly, the remaining particles showed a c-di-GMP-free BcsA synthase and a more extended vestibule conformation (Figs. 1c and 5a–c and Supplementary Fig. 2), where the central BcsE$^{REC*}$ domains engage in a different dimerization interface mediated by the pairs of β1–β2 connecting loops (melted α1 relative to canonical response regulators) and the C-proximal α-helices (canonical α5) (Fig. 5d). Densities for the PilZ-proximal GGDEF* domain feature markedly lower-resolution (Supplementary Fig. 8), however, the conformation for both BcsE$^{REC*-GGDEF*}$ tandems is still consistent with the closed BcsE state and intercalated c-di-GMP complexation (Fig. 5c, e and Supplementary Fig. 2d). The overall BcsE fold features a more extended conformation along the NTPase*-REC* domain linkers, neither BcsE protomer contacts the cytosolic synthase modules and the X-shaped BcsF dimer is found shifted near the fifth and fourth BcsB protomer as opposed to the c-di-GMP-saturated complex shown above (Fig. 1c). Nevertheless, the BcsRQ tandem is retained as an apical

complex and the synthase-proximal BcsQ and BcsR protomers engage in similarly extensive contacts with the catalytic and PilZ modules (Fig. 5f). The latter is only partially rotated around the hinge helix relative to the c-di-GMP-bound state (12.8° rotation and 1.7 Å displacement) and the N-proximal PilZ domain linker is partially unstructured but remains far from gating loop-stabilizing interactions with the BcsA$^{GT}$ core. Conversely, the gating loop remains unresolved, and the active site appears substrate-accessible, suggesting an overall preserved catalytically competent state in the assembled macrocomplex (Fig. 5g).

Together, these data suggest that even lower, non-saturating c-di-GMP concentrations would allow dinucleotide binding to the nanomolar-affinity sensor BcsE via contributions of both its REC* and GGDEF* domain I-sites and would lead to sufficient BcsE compaction, assembly of the cytosolic vestibule and stabilization of the synthase modules in a BcsRQ-preactivated state. The specific BcsE REC* domain dimerization interface and overall vestibule conformation would also be likely influenced by the lateral diffusion and BcsF partner stabilization among the synthase-distal BcsB copies of the crown. Processive substrate addition and product release by BcsA would thus depend

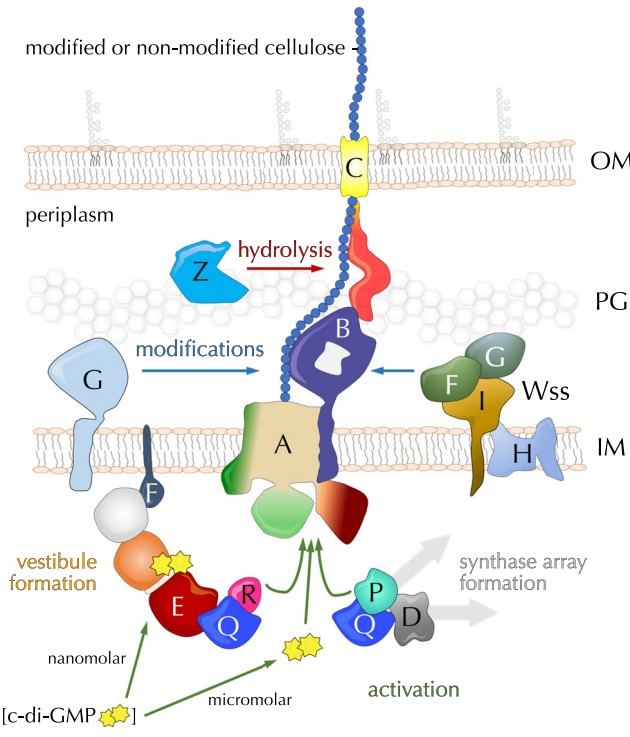

**Fig. 6 | Synthase activation and polymer modifications in β- and γ-Proteobacteria.** In addition to direct c-di-GMP complexation at micromolar dinucleotide concentrations, BcsA can be activated or stabilized in a catalytically competent conformation by a high-affinity c-di-GMP-sensing BcsRQEF cytosolic vestibule complex or by macromolecular intracellular scaffolds. In the periplasm, the polymer can undergo chemical modifications by the pEtN-transferase BcsG or by a multicomponent Wss cellulose acetylation complex. Finally, the polymer can undergo limited hydrolysis by the periplasmic endoglucanase BcsZ. OM outer membrane, PG peptidoglycan, IM inner membrane.

primarily on minute movements of its gating loop and finger helix—as observed in crystallo in saturating dinucleotide concentrations for the catalytic cycle of the *R. sphaeroides* BcsAB tandem—rather than be absolutely dependent on direct synthase-c-di-GMP complex formation. Overall, this is consistent with a model where the secondary c-di-GMP sensor BcsE serves as a proxy for dinucleotide-dependent regulation by effectively lowering the threshold for activating c-di-GMP concentrations and stabilizing the catalytically competent synthase state, rather than by circulating dinucleotide in and out of its PilZ-linker pocket (Fig. 6).

## Discussion

In many free-living and eukaryotic host-associated bacterial species, secreted cellulosic polymers represent key building components in the three-dimensional architecture of collaborative multicellular biofilms[3,6]. In *E. coli*, the pEtN decoration of secreted cellulose influences not only the physicochemical properties of the polymer itself, but also favors higher-order, long-range fibrillation of the other major biofilm matrix component—amyloid curli—and thus provides for markedly increased biofilm cohesion and elasticity[12,44]. Importantly, the mature biofilm is a highly heterogeneous environment with stark gradients of oxygen, nutrients, moisture, and/or shear stress. This leads to local stratification and/or compartmentalization of the quantity and type of secreted adherence factors and yields a self-organized division of labor where subsets of cells engage in extracellular matrix

production, while others provide for cell proliferation and/or biofilm dispersal[3,45].

In general, younger, nutrient-exposed biofilm layers are characterized by post-exponential growth metabolism, rod shape, enmeshed or no flagella, preserved proliferation, and very low c-di-GMP levels (~40–80 nM)[3]. Conversely, older, stationary-phase biofilm strata activate a cascade of c-di-GMP-metabolizing enzymes for gradual c-di-GMP increase, thus leading to non-dividing rounded cells embedded in a dense extracellular mesh of pEtN-cellulose and amyloid curli[3]. In intermediate layers, separate pockets or pillars of cells can activate specifically curli or pEtN-cellulose secretion, suggesting localized regulatory events that can selectively override the global c-di-GMP deficit[3]. Indeed, at least in some *E. coli* strains, the Bcs macrocomplex has been shown to directly interact with a cellulose-specific diguanylate cyclase (DgcC/AdrA), which would dramatically increase the probability of c-di-GMP-BcsA encounters in comparison to dinucleotide diffusion from an overall depleted cytosolic pool[43]. In addition to the spatial sequestration of a pathway-specific diguanylate cyclase and in light of the structural data presented here, we propose that *E. coli* and related enterobacteria have evolved a highly cooperative nanomachine for efficient c-di-GMP-sensing, cellulose synthase activation, and polymer modifications.

Consistent with our earlier but indirect BACTH results[17], we reveal here that the *E. coli*-like BcsA synthases have evolved a specific N-terminal amphipathic domain, whose W-shaped fold recruits three copies of the BcsG pEtN-transferase. The latter is an enzyme that is proposed to use inner membrane PE as a pEtN donor and to transfer the moiety via a S[278]-linked covalent intermediate; its catalytic domain and enzymatic mechanism have been extensively characterized structurally, in vitro, and in vivo[19,26,28,46]. It is important to note that whereas up to half of the glucose residues can be pEtN-modified in the processively secreted cellulose[19], PE is a small-headgroup zwitterionic phospholipid that is generally enriched in the inner, rather than the outer, leaflet of the inner membrane[47]. The evolution of enterobacterial BcsA[NTD] for the recruitment of multiple BcsG copies per synthase could thus provide efficient substrate mining for extensive polymer modification during processive synthase activity, where individual BcsG protomers are likely to act independently of each other. The membrane sampling and significant conformational changes, which would be required for pEtN-transfer onto a nascent polymer processively extruded through the periplasmic BcsB crown, are possibly enabled by the 48 amino-acid long interdomain linker that could at least theoretically extend more than 10–15 nm in the periplasmic space. Highly dynamic, large-scale structural transitions have been proposed based on molecular dynamics simulations for other pEtN-transferases, such as the lipid A pEtN-transferase *Nm*EptA[29], however, in the latter both the pEtN donor and acceptor (lipid A) are expected to be still embedded in the inner membrane. Further mechanistic work is thus necessary to capture substrate-, intermediate- and product-bound states across the catalytic cycle of full-length BcsG in the context of the multicomponent secretory assembly and translocating cellulosic polymer.

In addition to recruitment of the BcsG complex, we further reveal the recruitment and interactions of the rest of the *E. coli*-characteristic Bcs subunits, which are either essential for (e.g., BcsRQ) or greatly affect cellulose biogenesis (BcsEF) in vivo[17]. We previously demonstrated that, in the absence of BcsEFG or BcsE[NTD], BcsRQ are not stably retained in the macrocomplex, and the periplasmic crown features a pentameric, rather than hexameric BcsB[17,20]. In contrast, we reveal here that the assembly of a wild-type macrocomplex and a hexameric BcsB crown likely contributes not only the stabilization of the trimeric pEtN-transferase complex between BcsA[TMD] on one side, the synthase-distal BcsB copy on the other and BcsA[NTD] at the inner membrane-cytosol interface; but also to the recruitment of dimeric BcsF via discreet interactions with the

C-terminal tail-anchors of synthase-distal BcsB copies from the crown. We further demonstrate a cytosolic, BcsF-dependent β-sheet complementation mechanism for recruitment of the catalytically incompetent NTPase-like domain of BcsE, which itself leads to the stabilization of the entire $BcsE_2R_2Q_2$ cytosolic vestibule around the catalytic and c-di-GMP-sensing modules of the synthase. Although this vestibule is observed in two discreet c-di-GMP-bound conformations dependent on dinucleotide abundance and stabilizing BcsB-BcsF interactions, BcsA remains BcsRQ-bound and, as a result, presents a catalytically competent conformation even in the absence of direct dinucleotide complexation.

Together, these data highlight the possibility of two additional regulatory inputs for efficient synthase activation, which could have widespread implications across enterobacteria and beyond. On the one hand, the nanomolar-affinity, tandem I-site-presenting c-di-GMP sensor BcsE could effectively lower the threshold for activating c-di-GMP concentrations in the assembled Bcs macrocomplex (Fig. 6). Such activation by a separate c-di-GMP sensor is reminiscent of other widespread EPS secretion systems where activating c-di-GMP-sensing is carried out either with the contributions of (e.g., in the poly-*N*-acetylglucosamine secretion system of *E. coli*) or fully by separate co-polymerase subunits (e.g., in the Pel or alginate secretion systems of *P. aeruginosa*)[4]. In *E. coli* and other related bacteria, such activation-by-proxy could provide an important boost to cellulose secretion in the early stages and/or intermediate layers of biofilm development where cytosolic c-di-GMP is particularly low[3,5] and where functional differentiation between cell proliferation vs. biofilm matrix secretion provides the foundations of the three-dimensional matrix architecture without inhibiting overall macrocolony growth. On the other, the observed PilZ domain-stabilizing BcsA-BcsR interactions are likely preserved in a wide range of BcsP-encoding Bcs secretion systems that do not necessarily feature a *bcsEFG* cluster but could rather rely on BcsA-interacting BcsPDQ scaffolds for stabilization of the catalytically competent synthase state and enhanced polymer secretion[39] (Fig. 6 and Supplementary Fig. 1b). Both the more widespread and the idiosyncratic BcsA-regulatory mechanisms presented here can thus be harnessed for the selective targeting of a variety of cellulose secretion systems across free-living, pathogenic and symbiotic bacteria, as well as for the bioengineering of hybrid systems for the enhanced production of biotechnologically relevant polymers.

## Methods

No statistical methods were used to predetermine the sample size. The experiments were not randomized, and the investigators were not blinded during experimental design, execution, or outcome assessment.

### Bacterial strains and plasmids

Oligonucleotides, construct design, and bacterial strains are listed in Tables S1 and S2. All plasmids for recombinant protein expression (see below) were propagated in and isolated from *E. coli* DH5α cells. Recombinant Bcs macrocomplex expression for structural studies was carried out in NiCo21(DE3) competent *E. coli* cells (New England Biolabs). Recombinant expression for assessment of BcsF roles in macrocomplex assembly was carried out in a T1 phage-resistant Δ*bcs* BL21*(DE3) strain, featuring a deletion of both *bcs* operons (*bcsEFG* and *bcsRQAB*), as well as the corresponding interoperon region (see below). Phenotypic assays of colony morphology and calcofluor binding were carried out in the wild-type *E. coli* 1094 strain, and the *E. coli* 1094 Δ*bcsR* strains were transformed with variants of the low-copy pAM-238 plasmid. All bacterial strains and plasmids used in this study are available upon request.

### Recombinant DNA techniques

DNA manipulations were carried out using standard protocols for polymerase chain reaction (PCR), molecular cloning, transformation, and DNA analysis. Procedures for cloning of *bcs*^HIS^*RQA*^HA-FLAG^*B* and *bcs*^Strep^*EFG* for co-expression from pACYCDuet1 and pRSFDuet1* are similar to those previously described. Briefly, the genomic region corresponding to *bcsRQA*^HA-FLAG^*B* was amplified using genomic DNA from the *E. coli* 1094 *bcsA*^HA-FLAG^ strain as a template and a high-fidelity DNA polymerase (Phusion, New England Biolabs) with appropriate restriction sites introduced in the 5′ primer overhangs (sense/antisense PstI/NotI). In parallel, the pACYCDuet1 vector was also PCR-amplified to include the respective restriction sites for in-frame ligation under the pACYCDuet1 Promoter 1, including the incorporation of an N-terminal polyhistidine tag-coding sequence on *bcsR*. The genomic region corresponding to *bcsEFG* was PCR-amplified with appropriate restriction sites introduced in the 5′ primer overhangs (sense/antisense BamHI/NotI), and the pRFSDuet1 vector was amplified to introduce the respective restriction sites for in-frame ligation under the pRSFDuet1 Promoter 1 and to remove the polyhistidine tag-coding sequence (*). All PCR products were subsequently digested with the respective restriction enzyme pair (New England Biolabs), gel-purified, ligated using T4 DNA ligase (New England Biolabs), transformed into chemically competent DH5α cells, and plated on LB agar plates containing an appropriate antibiotic (34 µg ml⁻¹ chloramphenicol and 40 µg ml⁻¹ kanamycin for the pACYCDuet1 and the pRSFDuet1 constructs, respectively). Single colonies were grown in 5 ml liquid LB medium at 37 °C overnight, and the plasmid DNA was extracted using NucleoSpin® Plasmid preparation kit according to the manufacturer's instructions (Macherey-Nagel). Positive clones were identified by restriction digestion and DNA sequencing. For introduction of a N-terminal STREP II tag-coding sequence in *bcsE*, the purified *bcsEFG*-pRSFDuet1* was inverse PCR-amplified with oligonucleotides including the epitope tag-coding sequence, the PCR product was gel-purified, 5′ phosphorylated using T4 polynucleotide kinase (New England Biolabs), ligated by the addition of T4 DNA ligase and transformed in *E. coli* DH5α cells for plasmid selection and amplification as above[21].

### *E. coli* Δ*bcs* strain construction

The BL21*(DE3) Δ*bcs* mutant was generated using a modified protocol of a one-step inactivation procedure[48]. First, an FLP recognition target sites (FRT)-flanked kanamycin resistance (Km^R) cassette was generated by PCR using the pKD4 plasmid as a template and a pair of oligonucleotides carrying 50-nucleotide extensions homologous to regions adjacent to the target *bcs* gene cluster. In parallel, BL21*(DE3) was transformed with the pKD46 plasmid, and transformants were selected on LB agar plates supplemented with 100 µg/ml ampicillin and grown at 30 °C. Of these, a single colony was grown in liquid LB at 30 °C, in the presence of ampicillin and 0.05% arabinose for induction of phage λ Red recombinase prior to chemically competent cell preparation. The PCR product was then transformed into the resulting BL21*(DE3) cells, and transformants were selected on LB agar plates supplemented with 40 µg ml⁻¹ kanamycin and grown at 37 °C, allowing for the loss of the pKD46 helper plasmid. Replacement of the *bcs* gene cluster by the kanamycin-resistance cassette was confirmed by colony PCR. The resulting Δ*bcs*::Km^R strain was then transformed with the pCP20 helper plasmid, encoding Flp recombinase, and transformants were selected on ampicillin (100 µg ml⁻¹), then incubated for 24 h at 30 °C to allow excision of the cassette by the expressed Flp recombinase. Plasmid pCP20 was then eliminated by growth at 37 °C in the absence of antibiotics, and the cells were verified for kanamycin and ampicillin sensitivity.

### Protein overexpression and purification

Overexpression of the Bcs macrocomplex was performed by co-expression of the pACYCDuet1-*bcs*^His^RQA^HA-FLAG^B and pRSFDuet1*-

*bcs*[Strep]*EFG* constructs in chemically competent NiCo21(DE3) cells and plated on LB agar plates with antibiotic concentrations reduced to two-thirds of the ones stated above. After overnight incubation of the plates at 37 °C, multiple colonies of the transformed NiCo21(DE3) cells were picked and grown together at 37 °C in antibiotics-supplemented terrific broth (TB) medium to optical density at 600 nanometers ($OD_{600}$) of 0.8–1.2, upon which the cultures were transferred to 17 °C and induced with 0.7 mM isopropyl-β-D-thiogalactopyranoside (IPTG, Neo Biotech) for 16 h or overnight. Cells were pelleted by centrifugation (5000 × *g*, 20 min, 4 °C) and the pellets were resuspended in ice-cold buffer A containing 20 mM HEPES pH 8.0, 120 mM NaCl, 10% glycerol, 5 mM $MgCl_2$, 10 μM adenosine-5′-[(β,γ)-methyleno]triphosphate (AppCp, Jena Bioscience), 2 μM cyclic diguanylate (c-di-GMP, Sigma-Aldrich), 250 μM cellobiose, 0.5 mg ml⁻¹ *Aspergillus niger* cellulase (Sigma-Aldrich), 100 μg ml⁻¹ lysozyme, and 1 tablet per 50 ml complete EDTA-free protease inhibitors (Roche). The cells were subsequently disrupted using an Emulsiflex-C5 high-pressure homogenizer (Avestin) and the lysates were pre-cleared by a low-speed centrifugation step (10,000 × *g*, 15 min, 4 °C). Membranes were pelleted by high-speed centrifugation using an SW 28 Ti or an SW 41 Ti Beckman rotor (26,500 rpm/126,000 × *g* or 38,000 rpm/247,000 × *g*, respectively, for 1 h at 4 °C) and resuspended in solubilization buffer containing all buffer A components except for lysozyme and cellulase, as well as a mix of detergents at the following final concentrations: 0.6% w/v digitonin (Sigma-Aldrich), 0.35% w/v n-dodecyl-β-D-maltopyranoside (anagrade β-DDM, Anatrace), and 0.45% w/v lauryl maltose neopentyl glycol (LM-NPG, Anatrace). After incubation for 90 min at 22 °C and under mild agitation, the solubilized membrane fraction was cleared by a second high-speed centrifugation (50,000 × *g*, 40 min, 4 °C). The supernatant was incubated with ANTI-FLAG® M2 affinity gel (100 μl resin per litre of induced culture, Sigma-Aldrich), under mild agitation at 4 °C for 1 h. After gravity elution of the non-bound fraction, the resin was washed extensively (>30 column bed volumes) with affinity buffer containing 20 mM HEPES pH 8.0, 120 mM NaCl, 5 mM $MgCl_2$, 10 μM AppCp, 4 μM c-di-GMP, 250 μM cellobiose and 0.01% w/v LM-NPG. The bound complexes were eluted using four-column bed volumes of elution buffer (affinity buffer supplemented with 3× FLAG® peptide at 100 μg ml⁻¹), concentrated on a 100 kDa cutoff Amicon® Ultra (MerckMillipore) centrifugal filter. Samples were analyzed by SDS-PAGE and western blots. For cryo-EM grid preparation, the Bcs macrocomplex was concentrated to ~2–4 mg ml⁻¹, spotted on glow-discharged (ELMO, Cordouan Technologies) gold UltrAuFoil R 1.2/1.3 cryogrids, blotted, and plunge-frozen in liquid ethane using a Vitrobot Mark IV device (Thermo Fisher Scientific) at 4 °C and 100% humidity.

## Cryo-electron microscopy and single-particle analysis

Cryogrids were prescreened and optimized on the Elsa Talos Arctica transmission electron microscope (Thermo Fisher Scientific) at the European Institute of Chemistry and Biology (IECB, Bordeaux) operated at 200 kV and equipped with a Gatan K2 Summit direct electron detector. For structure resolution, cryo-EM data was collected at the CM01 beamline at the European Synchrotron Radiation Facility (ESRF, Grenoble) on a Titan Krios transmission cryo-electron microscope, operated at 300 kV and equipped with a GATAN K3 direct electron detector and a Quantum LS imaging filter. 20,022 movies (two movies per grid hole, 50 frames per movie) were recorded in electron counting mode with a total electron dose per movie of 49.35 electrons/Å², corrected pixel size of 0.839 Å/pixel, and defocus spread from −2.1 to −0.3 μm. The movies were motion-corrected using MotionCor2[49] within the ESRF autoprocessing pipeline and the resulting micrographs were imported in CryoSPARC[50] v4.4.1 for Patch-CTF correction and downstream processing. Particles were autopicked using the software's Template Picker function and 2D templates as previously reported[21] and, after

extraction (box size 500 pixels, Fourier crop 200) and a round of 2D classification, a total of 1,359,795 particles with resolved structural features were selected for further processing. Ab-Initio Reconstruction and Heterogeneous Refinement among three classes yielded a model consistent with the previously reported Bcs macrocomplex structure integrating 834,077 or 61% of the preselected particles. The corresponding particles were re-extracted without downsampling, and non-uniform refinement led to a 3D reconstruction featuring well-resolved crown densities and less-resolved inner membrane and cytosolic regions. The hexameric BcsB periplasmic crown was locally refined after subtracting the inner membrane and cytosolic densities from the particles dataset using the Particle Subtraction function. An inverse Particle Subtraction was also used to subtract the periplasmic densities from the initial particles dataset in order to retain only the inner membrane and cytosolic regions. The latter subtracted particles were then subject to another round of Ab-Initio Reconstruction with three classes yielding two well-resolved classes corresponding to the c-di-GMP-bound and the c-di-GMP-free synthase, whereas a third class featured poorly resolved structural features. Each of the resulting classes was input as a search model for heterogeneous refinement (3D classification) of the full macrocomplex, yielding the two states−c-di-GMP-saturated or not−for the global assembly. Corresponding particles were subject to another round of Ab-Initio modeling or each 3D class, followed by resolution-limited non-uniform refinement to avoid oversharpening and loss of the more dynamic/less resolved features. The respective crown regions were subtracted again, and separate regions of interest were further refined via Local Refinement jobs after map segmentation and mask generation within Chimera[51]. Additional map sharpening for density interpretation was performed using Deep EMhancer[52] via the CryoSPARC interface. Atomic model building and refinements were performed iteratively using previously reported BcsB, BcsRQ, and BcsE[REC*-GGDEF*] structures[20,21] and AlphaFold3[53] or ColabFold[54]-generated models as inputs for manual building in Coot[55] and automated real-space refinement in Phenix[56]. Interface analyses were carried out with the PISA server[57]. Details of the data collection and refinement statistics are listed in Tables S3 and S4, and Supplementary Figs. 3, 4, 8, and 9. Structure visualization was performed in ChimeraX[58].

## SDS-PAGE and western blot analyses

Protein fractions were analyzed by standard denaturing SDS-PAGE electrophoresis using 4–20% gradient mini-gels (Bio-Rad), InstantBlue Coomassie protein stain (Abcam), and a Bio-Rad GelDoc Go Infinity imager. For Western blot analyses, SDS-PAGE−migrated proteins were directly transferred using a standard mini-gel transfer protocol, polyvinylidene difluoride membranes, and a Trans-blot Turbo transfer system (Bio-Rad). Blocking and antibody incubations were performed in the presence of 5% skim milk or bovine serum albumin (the latter for STREP II tag detection) in TPBS (1× phosphate-buffered saline supplemented with 0.1% Tween-20 detergent); all washes between and after antibody incubations were performed with 1× TPBS buffer. Mouse anti-HA (hemagglutinin) (Thermo Fisher Scientific, #26183; dilution 1:1000) and mouse anti-STREP II (QIAGEN, #34850; dilution 1:1000) antibodies were used as primary antibodies; horseradish peroxidase-conjugated rabbit anti-mouse antibody (Abcam, ab6728; dilution 1:10,000) was used as secondary antibody. Signals were visualized using the Clarity Western ECL substrate and a ChemiDoc imaging system (Bio-Rad).

## Consensus structures modeling

BcsA, BcsP, BcsE, and BcsF protein sequences encoded by operons coding for BcsR-BcsE-BcsF (Type II *bcs* clusters, 20 representative sequences for each protein), BcsP-BcsD-BcsQ (Type I and hybrid *bcs* clusters, 30 representative sequences for each protein) or BcsK (Type III *bcs* clusters, 121 representative sequences for BcsA) were

identified with the help of webFlaGs[59] and the STRING[60] and NCBI Nucleotide databases and aligned separately using Clustal Omega[61]. The alignments were visualized in JalView[62] and trimmed for non-conserved N- or C-terminal extensions and internal sequence gaps. The corresponding consensus sequences were then retrieved, and the proteins or protein complexes were modeled using the AlphaFold[53] or ColabFold[54] web server and visualized in ChimeraX[58].

## Calcofluor- and congo red-binding assays

To test for the functional effects of the BcsA-interacting BcsR region, chemically competent cells were prepared from *E. coli* 1094 wild-type and Δ*bcsR* deletion strains. The latter was transformed with a low-copy-number plasmid (pAM-238) carrying none, wild-type or mutant *bcsR* genes and plated on LB agar plates (Miller) supplemented with 60 μg ml$^{-1}$ streptomycin. Single colonies were inoculated in 3 ml LB-streptomycin medium and left to grow overnight at 37 °C with agitation. On the following morning, 4 μl of each culture was spotted onto low-salt LB agar plates (1.5 g L$^{-1}$ NaCl) supplemented with streptomycin, 0.1 mM IPTG, and 0.02% calcofluor (fluorescent brightener 28; Sigma-Aldrich) or 25 μg ml$^{-1}$ Congo Red (Sigma-Aldrich). The spots were allowed to air dry, and the plates were incubated at 30 °C. After 24 h, the plates were photographed under brief illumination with long-wave UV light (365 nm) for calcofluor fluorescence and with a GelDoc Go imaging system (Bio-Rad) under trans-UVB illumination (UV tray and ethidium bromide mode) for pEtN-cellulose-specific Congo Red fluorescence.

## Reporting summary

Further information on research design is available in the Nature Portfolio Reporting Summary linked to this article.

## Data availability

All data needed to evaluate the conclusions in the paper are present in the paper and/or the Supplementary Information. Refined structural models and electron density maps are deposited in the electron microscopy and protein databanks with accession codes as follows: EMD-50584 and EMD-50595 for the low-pass filtered global assemblies of the c-di-GMP-saturated and non-saturated Bcs macrocomplex, respectively; 9FMT/EMD-50567 for the locally refined BcsB periplasmic crown; 9FMZ/EMD-50581 and 9FMV/EMD-50571 for the locally refined c-di-GMP-bound and c-di-GMP-free BcsAG3 complex, respectively; 9FNN/EMD-50599 and 9FP0/EMD-50632 for the locally refined crownless the c-di-GMP-saturated and non-saturated Bcs macrocomplex, respectively; 9FO7/EMD-50619 for the locally refined BcsE$_2$F$_2$ regulatory subcomplex from the c-di-GMP-saturated state; and 9FP2/EMD-50633 for the locally refined BcsRQEF vestibule complex from the non-saturated Bcs macro complex. Previously published structural models discussed in this work refer to entries 6YB3 (crystal structure of *E. coli* BcsRQ), 6TJ0 (crystal structure of splayed BcsE), 6YBB (crystal structure in closed BcsE, in a BcsRQ-bound complex), 6PCZ (a BcsG$^{CTD}$ crystal structure), 5FGN (crystal structure of *N. meningitidis* EptA), 6WLB (cryo-EM structure of poplar CesA8), 4P00 (a crystal structure of *R. sphaeroides* BcsAB). AlphaFold and ColabFold-generated models used in initial model building or structure analyses are deposited as an open-access dataset in Zenodo (DOI:10.5281/zenodo.13732043).

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

## Acknowledgements

We would like to thank current and former members of the Structural Biology of Biofilms group, and especially W. Abidi, A. Siroy, and M. Decossas, for discussions, technical assistance, and/or work peripheral to the project. This project has also benefited from and has directly contributed to the IECB cryo-EM platform, and we are grateful to R. Linares and E. Kandiah for data collection assistance at the CM01 beamline at the ESRF (Grenoble, France) and to E. Cascales and J.M. Ghigo for plasmid and strain sharing. This research has received funding from the ERC Executive Agency under grant agreement 757507 BioMatrix-ERC-2017-StG (to P.V.K.) and the *Agence Nationale de Recherche* (ANR, France) under grant agreements CelluSec (to P.V.K.) and T-ERC CoG 2024 BacFilm (to P.V.K.). Finally, Itxaso Anso is further supported by the Postdoctoral Program under the Order of 20 June

2023 of the Ministry of Education (Basque Country, Spain), which regulates and coordinates new grants and grant renewals for the Advancement of Doctorate-level Investigators (*Programa Posdoctoral de Perfeccionamiento de Personal Investigador Doctor*; to I.A.).

## Author contributions

P.V.K. conceived the project. I.A., S.Z., T.S., and P.V.K. designed, performed, and optimized the experimental procedures. I.A., S.Z., T.S., and P.V.K. analyzed the data. I.A. and P.V.K. secured funding and wrote the paper.

## Competing interests

The authors declare no competing interests.
