## [Peer Review File · Nature Communications]

Structural basis for synthase activation and cellulose modification in the E. coli Type II Bcs secretion systemReviewer #1 (Remarks to the Author):

In this manuscript, Krasteva et al. report the first high-resolution cryo-EM structure of the full inner membrane E.coli Bcs secretion system at local resolutions supporting side-chain modeling and resultant interpretations in most parts of the complex. The data presented in the manuscript is a vast improvement over previously reported low-resolution maps of the Bcs macrocomplex: its stoichiometry, as well as the domain interfaces supporting its architecture are thoroughly described and relevant references pointing to enzymology data validating the structural observations in this study are well integrated in the narrative. The inspection of the readily provided structure files (overall complex and locally refined substructures) confirms the robustness of the main data package supporting this manuscript and its interpretations. Structural and functional hypothesis for the decoration of neosynthesized cellulose by the phosphoethanolamine transferase BcsG, as well as an activation by proxy mechanism supported by higher-affinity binding of c-di-GMP by BcsE are also convincingly proposed.

Overall, I have enjoyed reading this paper, and it is obvious the corresponding author is well acquainted with this complex, having published several structure-function papers of most of its individual components. Perhaps it is therefore fortunate that I as a reviewer was unaware of the structural underpinnings of cellulose secretion, so that I could provide an uneducated view of how this manuscript is presented to the wide audience of this journal. I have not found anything precluding the publication of this piece, technical or argumentative, and these findings are very relevant to explain years of research on the structure of the Bcs complex. I do however have several suggestions on how to improve the understanding of what is a very complex and information rich piece for the naïve reader.

General

1) In your structure figures, I would recommend the use of labels clearly identifying the inner membrane and cytosol/periplasm to orient the reader.

2) I usually find the 3-letter amino acid code more legible when employed in the text. For example: "P43" -> "Pro43", "S278-linked covalent intermediate" -> "Ser278-linked covalent intermediate", with the exception of common nomenclature, e.g "R.D.R triad".

3) The authors often make mention of "domain X stabilizing domain Y". I understand the meaning to be that in the presence of domain X, domain Y that is normally disordered is resolved in the electron density. Stability can refer to a thermodynamic state that can be experimentally measured: does the BcsB crown really stabilize the BcsG trimer and BcsF dimer, or reduce their conformational heterogeneity by interacting with them? In practical words, would BcsG have a higher melting temperature as part of the complex than separately? Ok, this is probably nitpicking, but this nomenclature may trigger some biophysicists. Admittedly, the use of the word in this context is common in the cryo-EM field because the data is often anisotropic, but I thought I would point it out.

4) "Importantly, whereas one of the BcsR protomers is solvent-exposed and features an unstructured N-terminal domain, the other BcsR copy also interacts with the back of the BcsA catalytic module (Fig. 4a-c). Contrary to the crystallized states where BcsRNTD adopts a b-hairpin conformation, here residues D21-S30 fold into an N-proximal α -helix (α N) that U-turns into an extended linker before adopting the V-shaped C-terminal domain onto the BcsQ dimer interface (Fig. 4c)."

Is the solvent-exposed N-terminus of one of the protomers unstructured, or so disordered as to be invisible in the electron density? Has it been proven that secondary structure is formed upon

interacting with BcsQ?

Introduction

- 1) The introduction makes no mention of bacterial biofilm's relevance in the colonization of hosts in the medical field, or give examples of biotechnological applications. It may help framing the topic and outlining the impact of the study's findings to spend a few sentences on this.
- 2) I have found that the first two paragraphs of the discussion would actually fit better in the introduction to explain upfront that differences in c-di-GMP levels exists in the various layers of the biofilm. The understanding of why having a more sensitive sensor is useful to the bacterium then comes naturally.
- 3) Does the BcsA membrane transport domain really correspond to the TMD acronym (Trans-Membrane Domain)? I have not heard the terminology "conserved across kingdoms" before in this context. Is it meant to convey that BcsA is conserved in Archaeobacteria as well?
- 4) A supplementary figure depicting the architecture of type I/II/III Bcs systems with annotations for the subunit names and importantly their proposed function would help the naïve reader assimilate the information that is coming up in the results section. I found myself constantly going back to various parts of the paper to recall which function is assigned to which subunit, having it all in one place would be helpful.

Material and methods

- 1) I find it better to express centrifugation speeds for membrane preps in unit of gravitational constant; e.g 100000 g is the gold standard for membrane pelleting and reproducible with any UC rotor with a simple conversion from RPM.

Results

- 1) I find Figure 1 important as it sets the stage for the whole results section. A few suggestions:
 - a) Fig1a: BcsA NTD could be colored in green in the operon box.
 - b) Fig1a: collecting at the dissociation constant to capture both states in one dataset is clever! In this representation it wasn't immediately clear to me which state of the complex each Kd represents, maybe placing Kd ~ 2.5 μ M right under 'splayed' and Kd ~ 140 nM under 'closed' work better (please change 'nm' to 'nM').
 - c) Fig 1b: is it possible to highlight the c-di-GMP binding sites? I know there is not much space, but it would help understand upcoming figures. You could also consider moving the stoichiometry in the header and highlighting it better in bold or italics: c-di-GMP-saturation Bcs macrocomplex: BcsAB6... etc
 - d) I find that a direct comparison of saturated and non-saturated conformations is lacking. My mind is looking for a way to understand the large-scale differences triggered by high concentrations of c-di-GMP. Could it be done by showing the two complexes side by side and with arrows? I imagine an overlay will be too messy, so how about a movie?

- 2) There is a lot of repetition between the last paragraph of the introduction and the first of the results that is used to set the stage ("Stoichiometry of the assembled Bcs macrocomplex of E. coli"). In my view, I would remove the former, and move the latter to the introduction, as it does a better job at summarizing the state of the art. From: "Here we present cryo-EM structures of the E. coli Bcs macrocomplex positioning all seven..." I suggest keeping it short and letting the upcoming results speak for themselves. Describing the complex in any level of detail prior to having seen and digested the large amount of information contained in these two structures is

quite overwhelming; probably not for scientists familiar with this complex, but they likely represent a tiny portion of the readers.

3) Regarding your hypothesis for the BcsG trimer function: in crystallography studies of EptA from *Alice Vrieling*, the linker forms an amphipathic helix and likely represents the enzyme in the resting state. Since the helix is amphipathic and there is very significant buried surface area between the TMD, linker and PEA transferase domains, it is actually not that clear to me that EptA can adopt the extended conformation you (and AlphaFold) depict here for BcsG. I would be careful here, as the large scale conformational changes you mention with regards to EptA are only a hypothesis borne of molecular dynamics. This assumption was never thoroughly demonstrated, and it is actually a disputed hypothesis, as it is generally not thermodynamically favorable for an enzyme to move that much unless it is absolutely necessary. Additionally, in the case of EptA both substrate and product are located closely to one another in the inner membrane, and a compact architecture would likely afford more efficiency in substrate recognition and PEA transfer. This is obviously not the case for BcsG; a different structural mechanism would likely be required because of the distances separating substrate and product (can this be estimated?). It would make sense for evolution to have selected a more flexible, unstructured linker for modification of the nascent cellulose. Can interdomain linker sequences and predicted secondary structure or lack thereof be compared against LPS modification EptA-like enzymes and others to explain this?

Discussion

- 1) I have fundamental questions regarding your activation by proxy hypothesis that may merit clarification in this section if the information is available:
 - a) Why evolve an additional subunit for c-di-GMP sensing, and not simply tune the affinity of the synthase binding site to detect lower amounts of nucleotide at all times?
 - b) Are there synthases without BcsE equivalent, or bound to an ortholog deficient for c-di-GMP binding? If so, have their catalytic efficiencies been compared in vitro at different concentrations of nucleotide?
 - c) The different architectures of the Bcs complex represented in Gram-negative bacteria suggest there is more than one solution to one problem. What gives rise to these different architectures, or how are they differentiated (nutrient abundance in the niche? Accessibility of cellulose building blocks?)?
 - d) Does activation of the synthase at lower c-di-GMP concentrations via the BcsE sensor yield similar catalytic efficiency (e.g. k_{cat}) as when micromolar concentrations of c-di-GMP are present? If yes, would it mean bacteria secrete cellulose at the same rate in different layers of the biofilm?

Reviewer #2 (Remarks to the Author):

Summary Comments to the authors:

The submitted manuscript by Anso et al. provides new details regarding the structural arrangement, stoichiometry and conformations of BcsF/E/R/Q subunits within the cellulose synthase macrocomplex. These subunits are also examined in light of the positioning of BcsA/B/G at the membrane interface and in many cases directly corroborate previously

published indirect data and match presumed biological capacities. This work combines structural analysis by a number of means (cryo-EM, AlphaFold2 and crystal structure data manipulation) and is a remarkable feat, as it delineates for the first time important protein-protein interaction surfaces in the presence and absence of the c-di-GMP molecule that is typically viewed as an ON/OFF switch, but now should be viewed more broadly in the context of other overriding activation factors (including different complex conformations) across different c-di-GMP landscapes that contribute to biofilm formation.

Main Comments:

The manuscript was well-written and provides a wealth of new information with regards to the cytosolic “vestibule” and its integration into the BcsAB complex. Importantly, the authors have recognized the presence and simultaneous submission of another manuscript that provides some overlapping structural data with regards to the cellulose synthase complex (i.e. reference 30 by Verma et al.). In many ways these manuscripts corroborate each other and these have been noted well in the present manuscript, but there are some subtle differences (ex. BcsB interactions) that could use some clarification/reasoning/insight to address differences in experimental data or interpretation. In all cases, these differences are understandable given the purification and analytical techniques used, but should still be noted for further work to clarify more fully rather than risk becoming dogma.

With regards to the BcsE conformations in BcsA bound/unbound c-di-GMP states, the authors focus on how this can allow polymer production even when c-di-GMP is absent from BcsA. This is still speculative and so the authors may want to be careful with how strongly they promote this. A more theoretical/suggestive viewpoint would be more pragmatic. Also, they fail to reference how this may be affecting BcsG activity. Given that BcsEFG are all part of the same operon and are in proximity to each other physically in the membrane (BcsG/F), it is feasible that these c-di-GMP states could also be affecting modification events. Please comment on this possibility in the manuscript.

Other Specific Minor Comments:

Note1: the use of line numbers would have helped in the review/edit process.

Note2: Please check the figures and references for proper italics on bacterial names

Note3: “Vestibule” - Not sure this is the right word for this as it represents a room or hallway next to an entrance. The BCSEK complex is more like a doorman than a room, but I am not too fussed about this if the authors have a different interpretation and prefer the term.

1 – reword “host-associated microbes such as” to “host-associated microbes, such as”

2 - reword “transferase (GT) and an membrane transport” to “transferase (GT) and membrane transport”

3 – pg2./2nd paragraph- you define the different Bcs systems. It was noticed later on that figure 6 has the acetylation machinery in the model. Perhaps here you could also address this modification with a sentence or two and which type it fits into?

4 – pg.3/2nd paragraph – In terms of stoichiometry it seems that most of the Bcs components are in pairs, but here it is noted that BcsG is a trimer. What is the relevance of this to the symmetry in the complex?

5 – “Importantly, the higher-affinity BcsE sensor remains c-di-GMP-bound in both structures and retains a compact conformation 15 to retain the regulatory BcsR2 Q 2 complex onto the

synthase, likely preserving the latter's catalytically competent state." What about affects on BcsB or BcsG? Can you reword to be more encompassing?

6 – reword “has been shown the associate with” to “has been shown to associate with”

7 – pg.4/para3 - Figure 1 has individual coloring that should be referenced in brackets as you talk about the different components here. This will help the reader sift through the positioning of each player in the macrocomplex more readily.

8 – pg.5/para2 – “The first and second copies BcsB copies engage in contacts with the BcsA synthase”. Recent Verma et al. manuscript suggests 1 BcsB and BcsA interaction. Can you address this discrepancy?

9 – pg.7/para2 – “The cryo-EM structures presented here are consistent with the previously tripartite architecture of BcsE, comprising a degenerate trio of an NTPase*, REC* and GGDEF* domains (Fig. 1a)”. I would suggest moving this part of Figure 1 to Figure 3 to follow the order of the manuscript better and place it in better context.

10 - pg7 – reword “different orientations relative to the to the apical”

11 – pg8/para2 - This importantly extends this work and further defines the different types of cellulose synthases. While the presence of a hydrophilic pocket is mentioned here and sequence conservation, a reason or importance of this is not noted for the type III. Is this likely to be a protein-protein interaction domain? What would the purpose purportedly be? This area is nicely compared for type 2 and type 1, but is lacking for type 3.

12 – pg9/last para – reword “insights, we kept low micromolar concentration”

13 – pg10/para2 – “X-shaped BcsF dimer is found shifted between the fourth and fifth BcsB protomer as opposed to the c-di-GMP-saturated complex”. Is there interaction between F and G or E and G in this conformation too that may influence PEtN modification of cellulose? Could it uncouple BcsG activity from BcsA possibly? Given the influence of BcsB and BcsG on BcsA activity, this alternate conformation may have an unidentified effect that is propagated at the membrane level and not through RQ? Expanding on more of this (especially if there is structural evidence/variation in support) and what is briefly mentioned in the next paragraph would be good.

14 – pg11 – reword for better clarity “Conversely, older, stationary phase strata cascade-activate specific subsets”

15 – pg11/last para – “whose catalytic domain and enzymatic mechanism have been extensively characterized in vitro and in vivo^{13,21,23}”. This paper (PMID: 34762795) I also tend to miss, but probably has the best in vitro characterization and should likely be mentioned here if you want to be comprehensive.

16 – pg11/12 – “The membrane sampling and significant conformational changes, which would be required for pEtN-transfer onto a polymer processively extruded through the periplasmic BcsB crown,”. It might also be pertinent to mention that the presence of three BcsG could simply be to try to match the catalytic rates of cellulose synthesis by BcsA with PEtN modification by BcsG in vivo, which are most likely not the same in a 1:1 scenario. Precise PEtN levels might be important for biofilm formation and maintenance and so complementary rates between BcsA and BcsG to achieve 50% modification would likewise be important to control and achieved by simply having 3 copies of BcsG.

Figure 1

- reword “have remained largely unresolved in the, whereas crystallographic snapshots”

- 3rd panel of A - Unclear why we need the 90 degree rotation of BcsB here but nothing else?

Maybe just show E in the splayed and closed conformations? Or keep this for the later figure on

BcsE. Indeed it seems a bit out of place here in the manuscript.

- at first glance the coloring of E and A makes this a bit confusing to follow. May think about simplifying this to match the operon more?
- Panel B – Better labelling of the following would help: 1. Where is the membrane? 2. E is a bit confusing across the panels 3. Label G and B in more than first panel.
- Panel C – More labelling here would help. Also, these conformational changes with respect to activation and activity should be expanded on a bit more. The authors have done a good job of explaining how ERQ locks into a position for activity, but how this works relative to these large macrocomplex conformational changes is still not entirely clear in the body of the text.

Figure 2

- showing the location of the membrane in panel A would help for orientation

Figure 3

- The ranges (colors) of the electrostatic and hydrophobic surfaces should be noted for the reader for clarity of interpretation.
- Location of membrane in panel A

Figure 4

- Location of membrane in panels A and B

Figure 5

- Location of membrane in panels A, B and C

Figure 6

- “by the pEtN-transferase BcsG or by a multicomponent Wss cellulose acetylation complex”. As noted earlier, this is introduced out of nowhere at the very end. I like that this is being a bit more comprehensive with possibilities, but you might want to add this into the introduction and discussion somewhere too. Even just a sentence that it is a possibility that fits somehow into the Types of cellulose biosynthesis.

Figure S1

- Is there any evidence of BcsC interaction with BcsB?

Figure S4

- Indicate BcsE and BcsF coloring in the last two panels

Structural basis for synthase activation and cellulose modification in the *E. coli* Bcs secretion system

Itxaso ANSO^{1,2,3}, Samira ZOUHIR^{1,2,4}, Thibault G. SANA^{1,2} and Petya Violinova KRASTEVA^{1,2,*}

POINT-BY-POINT REPLY TO THE REVIEWERS

Reviewer #1 (Remarks to the Author):

In this manuscript, Krasteva et al. report the first high-resolution cryo-EM structure of the full inner membrane E.coli Bcs secretion system at local resolutions supporting side-chain modeling and resultant interpretations in most parts of the complex. The data presented in the manuscript is a vast improvement over previously reported low-resolution maps of the Bcs macrocomplex: its stoichiometry, as well as the domain interfaces supporting its architecture are thoroughly described and relevant references pointing to enzymology data validating the structural observations in this study are well integrated in the narrative. The inspection of the readily provided structure files (overall complex and locally refined substructures) confirms the robustness of the main data package supporting this manuscript and its interpretations. Structural and functional hypothesis for the decoration of neosynthesized cellulose by the phosphoethanolamine transferase BcsG, as well as an activation by proxy mechanism supported by higher-affinity binding of c-di-GMP by BcsE are also convincingly proposed.

Overall, I have enjoyed reading this paper, and it is obvious the corresponding author is well acquainted with this complex, having published several structure-function papers of most of its individual components. Perhaps it is therefore fortunate that I as a reviewer was unaware of the structural underpinnings of cellulose secretion, so that I could provide an uneducated view of how this manuscript is presented to the wide audience of this journal. I have not found anything precluding the publication of this piece, technical or argumentative, and these findings are very relevant to explain years of research on the structure of the Bcs complex. I do however have several suggestions on how to improve the understanding of what is a very complex and information rich piece for the naïve reader.

We thank the reviewer for this great summary and the positive evaluation of our work. We address the various specific comments below and thank the reviewer for her/his great attention to detail!

General

1) In your structure figures, I would recommend the use of labels clearly identifying the inner membrane and cytosol/periplasm to orient the reader.

We have added labels to the figure panels.

2) I usually find the 3-letter amino acid code more legible when employed in the text. For example: "P43" -> "Pro43", "S278-linked covalent intermediate" -> "Ser278-linked covalent intermediate", with the exception of common nomenclature, e.g. "R.D.R triad".

For the sake of space and label legibility, we have opted for the single-letter code in the figures. In addition, we feel that mixing single-letter and three-letter codes are actually more confusing as there are many longer amino acid motifs (e.g. L²⁵-F²⁹-L³¹-I³⁴, RxxD I-site, RDR triad, etc.) that can be difficult to follow if split

between lines in a three letter code. We have therefore stuck to unique representation in single-letter code.

3) The authors often make mention of “domain X stabilizing domain Y”. I understand the meaning to be that in the presence of domain X, domain Y that is normally disordered is resolved in the electron density. Stability can refer to a thermodynamic state that can be experimentally measured: does the BcsB crown really stabilize the BcsG trimer and BcsF dimer, or reduce their conformational heterogeneity by interacting with them? In practical words, would BcsG have a higher melting temperature as part of the complex than separately? Ok, this is probably nitpicking, but this nomenclature may trigger some biophysicists. Admittedly, the use of the word in this context is common in the cryo-EM field because the data is often anisotropic, but I thought I would point it out.

We cannot evaluate the effect on melting temperatures on individual subunits or binary complexes mainly because most of the Bcs components do not express or are unstable upon purification on their own, or do not adopt the same interfaces when purified separately. We have revised the use of the word stabilize and have either substituted it (e.g. with ‘positioned’ or ‘packs against’) or have kept it while clarifying that it is the conformation or interface that are stabilized by the specific interactions.

In contrast, we have previously demonstrated that BcsRQ stabilize and act as chaperones to each other at the protein folding level (Zouhir et al. *mBio* 2020). In particular, BcsR prevents aggregation of BcsQ and itself does not express stably in the absence of BcsQ, even if it can be isolated when expressed in the presence of certain BcsQ mutant variants (Zouhir et al. *mBio* 2020). The above points towards BcsQ and BcsR stabilizing each other and whereas BcsR^{CTD} is sufficient for a stable BcsRQ complex formation and is key for cellulose secretion, BcsR^{NTD} itself remains essential (Abidi et al. *Science Advances* 2021).

BcsRQ expression and their BcsE-dependent integration in the Bcs macrocomplex also has direct effects on BcsA stability, autoproteolysis and detectability in the membrane (Krasteva et al. *Nat Commun* 2017; Zouhir et al. *mBio* 2020, Abidi et al. *Science Advances* 2021). In this context we cannot exclude that stabilization goes beyond positioning but also in terms of overall stability/lifetime of the interacting proteins (e.g. BcsR, BcsQ and BcsA).

We have already included the appropriate citations but we have underlined that “upon co-expression BcsR and BcsQ stabilize and act as chaperones to each other via the formation of a heterotetrameric BcsR₂Q₂ complex²⁰ with essential roles in Bcs system positioning, assembly, stability and function^{17,21,39}” (lines 272-274).

4) “Importantly, whereas one of the BcsR protomers is solvent-exposed and features an unstructured N-terminal domain, the other BcsR copy also interacts with the back of the BcsA catalytic module (Fig. 4a-c). Contrary to the crystallized states where BcsRNTD adopts a b-hairpin conformation, here residues D21-S30 fold into an N-proximal α -helix (α N) that U-turns into an extended linker before adopting the V-shaped C-terminal domain onto the BcsQ dimer interface (Fig. 4c).” Is the solvent-exposed N-terminus of one of the protomers unstructured, or so disordered as to be invisible in the electron density? Has it been proven that secondary structure is formed upon interacting with BcsQ?

We have previously characterized BcsR stability and functional roles of its C- and N-terminal regions in cell-based phenotypic assays, co-purification studies and BcsRQ and BcsE*RQ crystallographic snapshots (Zouhir et al. *mBio* 2020 and Abidi et al. *Science Advances* 2021; BcsE*: REC*-GGDEF* or GGDEF* only). In most

crystallographic snapshots the N-terminal domain is either not resolved (i.e. conformationally disordered or heterogeneous) or forms a β -harpin threading onto the surface of a BcsQ partner or symmetry mate.

We have changed “disordered” to “unresolved” in line 286.

We have included a supplementary figure (now Supplementary Fig. 6) comparing the crystallographic BcsR^{NTD} conformation onto BcsQ and the BcsR^{NTD} fold onto the synthase as visualized in the cryo-EM structure(s) resolved here. Importantly, while the latter’s stabilizing I22:Y36 interaction is mediated by side chain distances smaller than 4 Å, these residues are more than 40 Å apart in the crystal structure. This has been underscored in the supplementary figure and the text.

These distances are also now included in the text (lines 292-294).

Introduction

1) The introduction makes no mention of bacterial biofilm’s relevance in the colonization of hosts in the medical field, or give examples of biotechnological applications. It may help framing the topic and outlining the impact of the study’s findings to spend a few sentences on this.

We have included a short paragraph to highlight the significance of bacterial cellulose secretion. In particular, we have highlighted beneficial or noxious aspects of various types of cellulosic polymers and even context-dependent properties of the same type of secreted cellulose (lines 51-65).

2) I have found that the first two paragraphs of the discussion would actually fit better in the introduction to explain upfront that differences in c-di-GMP levels exists in the various layers of the biofilm. The understanding of why having a more sensitive sensor is useful to the bacterium then comes naturally.

We thank the reviewer but after trying to restructure the manuscript, we respectfully disagree.

Adding these paragraphs to the introduction makes it very long and we find that the addition of more conceptual points would actually distract the reader without prior information on the relevance of c-di-GMP levels to the results. This is especially so since we already extended the introduction to highlight cellulose secretion significance and removed the results summary that can put c-di-GMP levels in context (see detailed reply to Comment/Results/2)).

We therefore believe that biofilm stratification and differences of c-di-GMP levels become relevant and central in light of the new experimental data and therefore highlighting them is suited best in the discussion rather than the introduction.

We have thus preserved the original structure of the manuscript and have kept the introductory parts focused on the various Bcs components which are needed to understand the complexity and multiple functionalities within the system.

3) Does the BcsA membrane transport domain really correspond to the TMD acronym (Trans-Membrane Domain)??

As cellulose synthases are not typically defined as transporters, we prefer to keep the domain nomenclature and abbreviations as in the rest of our body of work. So we keep the domain organization as BcsA^{NTD}, N-terminal domain; BcsA^{GT}, glycosyl transferase domain; BcsA^{TMD}, transmembrane/export domain; BcsA^{PilZ}, PilZ

domain; and BcsA^{CT}, C-terminal tail/amphipathic helices. A BcsA domain architecture thumbnail is added to Figure 1a and the abbreviations are annotated in the figure panels.

TMD is also used more generally for transmembrane domain in the case of BcsG^{TMD} as well.

I have not heard the terminology “conserved across kingdoms” before in this context. Is it meant to convey that BcsA is conserved in Archaeobacteria as well?

We have analyzed previously similarities between the core folds of structurally characterized bacterial and plant cellulose synthases (Abidi *et al*, *FEMS Rev Microbiol* 2023), both likely evolved from ancestral cyanobacteria. We have added a supplementary figure panel to underscore the bacteria-plant cross-kingdom fold conservation here (now Supplementary Fig. 1a).

4) A supplementary figure depicting the architecture of type I/II/III Bcs systems with annotations for the subunit names and importantly their proposed function would help the naïve reader assimilate the information that is coming up in the results section. I found myself constantly going back to various parts of the paper to recall which function is assigned to which subunit, having it all in one place would be helpful.

We have included a supplementary figure panel (now Supplementary Fig. 1b) of various Bcs secretion systems across Gram-negative bacteria.

Material and methods

1) I find it better to express centrifugation speeds for membrane preps in unit of gravitational constant; e.g 100000 g is the gold standard for membrane pelleting and reproducible with any UC rotor with a simple conversion from RPM.

We have included the speed in RCF for the specific rotors, i.e. 126-247 kg (lines 576-577).

Results

1) I find Figure 1 important as it sets the stage for the whole results section. A few suggestions:

a) Fig1a: BcsA NTD could be colored in green in the operon box.

We have added a BcsA domain architecture thumbnail to figure 1a with color coding for all BcsA modules. We have also revised the figure legend to include the color-coding (lines 708-729)

b) Fig1a: collecting at the dissociation constant to capture both states in one dataset is clever! In this representation is wasn't immediately clear to me which state of the complex each Kd represents, maybe placing Kd ~ 2.5 uM right under 'splayed' and Kd ~ 140 nM under 'closed' work better (please change 'nm' to 'nM').

We have corrected 'nm' to 'nM', have recolored the arrow in the Kd values to match the conformational change arrow and have reshuffled the figure panels to position 'splayed' and 'closed' above the dissociation constants. We have also included the metrics of REC* domain displacement during the conformational rearrangement (also included in the figure legend, line 723).

c) Fig 1b: is it possible to highlight the c-di-GMP binding sites? I know there is not much space, but it would help understand upcoming figures. You could also consider moving the stoichiometry in the header and highlighting it better in bold or italics: c-di-GMP-saturation Bcs macrocomplex: BcsAB6... etc

We have increased the font of the respective stoichiometries. Highlighting the c-di-GMP-binding sites however is difficult in this context and we have done so in the subsequent subcomplex figures. We have however added additional figure panels to the supplementary figure (now Supplementary Fig. 1) to highlight the over differences in conformation, the specific differences in the regulatory vestibule and to indicate the additional c-di-GMP-binding sites on BcsE in both conformations.

d) I find that a direct comparison of saturated and non-saturated conformations is lacking. My mind is looking for a way to understand the large-scale differences triggered by high concentrations of c-di-GMP. Could it be done by showing the two complexes side by side and with arrows? I imagine an overlay will be too messy, so how about a movie?

It is unfortunately very complex to overlay the two structures as there are simply too many elements in each complex and we have not found a representation scheme that is not even more jarring. Instead we have expanded the supplementary figure. We have added an overlay of the lowpass-filtered full complex density reconstructions to the supplementary data and have included close-ups of the vestibule conformations in both electron density and cartoon representations (Supplementary Fig. 2c-d).

Regarding Fig. 1b-c, we have aimed to underscore the major conformational changes by keeping the color code, as well as the relative orientation and discrete rotations among the two complexes as close as possible. The specific orientations of functionally important modules are then underscored in the respective structures from local refinements through the rest of the manuscript.

2) There is a lot of repetition between the last paragraph of the introduction and the first of the results that is used to set the stage ("Stoichiometry of the assembled Bcs macrocomplex of E. coli").

In my view, I would remove the former, and move the latter to the introduction, as it does a better job at summarizing the state of the art. From: "Here we present cryo-EM structures of the E. coli Bcs macrocomplex positioning all seven..." I suggest keeping it short and letting the upcoming results speak for themselves. Describing the complex in any level of detail prior to having seen and digested the large amount of information contained in these two structures is quite overwhelming; probably not for scientists familiar with this complex, but they likely represent a tiny portion of the readers.

We agree with the reviewer that there is repetition between the introduction and the first section of the Results. We restructured the text, albeit slightly differently.

In particular, we have removed the detailed results summary from the introduction but have kept the Results as originally structured. This choice was made as the first Results section does not readily correspond to the state-of-the-art but rather reflects and summarizes actual findings of this work.

3) Regarding your hypothesis for the BcsG trimer function: in crystallography studies of EptA from Alice Vrieling, the linker forms an amphipathic helix and likely represents the enzyme in the resting state. Since the helix is amphipathic and there is very significant buried surface area between the TMD, linker and PEA transferase domains, it is actually not that clear to me that EptA can adopt the extended conformation you (and alphaFold) depict here for BcsG. I would be careful here, as the large scale conformational changes you mention with regards to EptA are only a hypothesis borne of molecular dynamics. This assumption was never thoroughly demonstrated, and it is actually a disputed hypothesis, as it is generally not thermodynamically favorable for an enzyme to move that much unless it is absolutely necessary.

Additionally, in the case of EptA both substrate and product are located closely to one another in the inner membrane, and a compact architecture would likely afford more efficiency in substrate recognition and PEA transfer. This is obviously not the case for BcsG; a different structural mechanism would likely be required because of the distances separating substrate and product (can this be estimated?). It would make sense for evolution to have selected a more flexible, unstructured linker for modification of the nascent cellulose. Can interdomain linker sequences and predicted secondary structure or lack thereof be compared against LPS modification EptA-like enzymes and others to explain this?

We have to underscore here that we do not know at which level the modification occurs and which state the default AlphaFold model, onto which we have overlayed the actual BcsG^{CTD} crystal structure, likely represents. It is more likely that the latter is a resting or pre-catalytic conformation where the active site is facing the inner membrane domain due to coevolving residues between the two folded modules. It is also to be remarked that phospholipid or partner-induced conformational changes and local folding of the interdomain linker cannot be excluded or deduced from simple modeling in AlphaFold and perhaps intermediate and product-bound states can be captured by the use of synthetic donor and product homologs in the future.

Regarding other EptA-like enzymes: In PbgA, an EptA/BcsG-like protein required for outer membrane integrity through lipopolysaccharide biogenesis, the interdomain linker adopts a more stable (even in molecular dynamics simulations) lipid A-binding fold and is referred to as an interfacial domain (Clairfeuille et al., Nature 2020). However PbgA is a catalytically inactive pseudohydrolase and, to our understanding, it likely regulates LPS biosynthesis by sensing excess LPS and regulating the stability of the cytosolic biosynthetic enzyme LpxC via indirect protein-protein interactions that provide a negative feedback loop (Clairfeuille et al., Nature 2020). Thus, to us it is not entirely surprising that the interdomain PbgA linker adopts a more stable ligand-sensing fold vs. a more dynamic conformation involved in substrate and product mining in a catalytically active enzyme.

It is also to be noted that - as we discussed in Abidi et al., FEMS Rev Microbiol 2023 - important aspects of BcsG function that have not yet been investigated are the substrate and byproduct of the enzymatic reaction itself, namely the need for a significant flux of phosphatidylethanolamine (PE) toward the outer leaflet of the inner membrane, as well as the release of a diacylglycerol (DAG) moiety for each pEtN addition onto the nascent polysaccharide.

Zwitterionic PE is the major phospholipid in bacteria, comprising ~75% of the phospholipid content in *E. coli* (Sohlenkamp and Geiger 2016). About half of it is found in the inner membrane, however with a significant asymmetry between its inner and outer leaflets (75% vs 25%, respectively, in rod-shaped bacteria) (Bogdanov et al. 2020). This asymmetric distribution is determined by the relatively small headgroup of the phospholipid and its preference for negative membrane curvature (Bogdanov et al. 2020). DAG, on the other hand, is a minor lipid in the bacterial envelope and generally a precursor for the synthesis of various phospholipids (Sohlenkamp and Geiger 2016). It has an even smaller head group than PE and a propensity to introduce localized hotspots of negative curvature and phase separations in the membrane, and in higher organisms can act as an important second messenger in a variety of cellular processes (Carrasco and Mérida 2007).

It is therefore intriguing that BcsG both uses and produces lipids that are not normally enriched in the outer leaflet of the inner membrane. This also adds to the nature of the pEtN-accepting cellulose, likely modified in the periplasmic space and not in immediate proximity to the membrane, as the reviewer also highlights.

In such context, it is possible that the protein requires more significant conformation variability for substrate mining, similar to what has been proposed for EptA (and regardless to whether the latter's movements actually occur *in vivo*) and in contrast to what appears to be a more static sensing in LPS-binding PbgA. The nature and

immediate availability/scarcity of BcsG substrate molecules can also explain the evolution of the BcsA N-terminal domain for the recruitment of multiple BcsG copies per single catalytically active and processive cellulose synthase for efficient polymer modification, where up to half of the glucose moieties can be modified (Thongsomboon et al. 2018).

Of course the above comparison between EptA, PbgA and BcsG are mostly hypothetical at this point and a bit off-focus, and the structures of a full-length PE-, pEtN- or pEtN-cellulose-bound BcsG remain to be characterized in future studies. We therefore believe that this discussion is better suited to this publicly available review document rather than further detailed in the body of the manuscript.

We have indicated that the EptA dynamic conformation is only 'proposed based on molecular dynamics situations' and that both the donor and acceptor (lipid A) are positioned in the inner membrane. We have also included the caveat that 'further mechanistic work is thus necessary to capture substrate-, intermediate- and product-bound states across the catalytic cycle of full-length BcsG in the context of the multicomponent secretory assembly and translocating cellulose' (lines 450-459). We have kept also short mention of PE substrate limitation as the potential requirement for multiple BcsG copies per synthase (lines 445-450).

Discussion

1) I have fundamental questions regarding your activation by proxy hypothesis that may merit clarification in this section if the information is available:

a) Why evolve an additional subunit for c-di-GMP sensing, and not simply tune the affinity of the synthase binding site to detect lower amounts of nucleotide at all times?

Not all cells in a biofilm secrete cellulose at a time as it is an energetically costly biosynthetic process that is more commonly associated with nutrient deprivation and inhibited cell proliferation. So at least conceptually it makes sense to introduce an additional regulatory input that can adapt to the different stages of biofilm formation and their respective cell metabolism, morphology and intracellular c-di-GMP levels that would provide an overall benefit to the multicellular life form.

It has been shown that cellulose secreting cells feature different shapes, metabolism and, most likely, c-di-GMP levels between biofilm layers: younger post-exponential-growth, low-c-di-GMP cells in cellulose-rich vertical pillars deep in the biofilm are surrounded by dividing cells devoid of biofilm matrix components, whereas cells in the upper and older biofilm layers are virtually coccoid, non-dividing and encased in individual baskets of biofilm polymers (Serra and Hengge, Annu Rev Micro 2021).

We have underscored that such stratification and functional differentiation likely provides balance between biofilm matrix secretion and proliferation and is necessary to secure the foundations for the 3D extracellular matrix without inhibiting the overall macrocolony growth (lines 421-424).

b) Are there synthases without BcsE equivalent, or bound to an ortholog deficient for c-di-GMP binding? If so, have their catalytic efficiencies been compared in vitro at different concentrations of nucleotide?

Whereas we and others have previously shown that the BcsAB tandem and the Bcs macrocomplex are catalytically active in detergent solubilized form, we have not characterized the catalytic efficiencies of the synthase itself and are planning to do so in future work in correlation with biofilm development studies. We have shown however that many hybrid and Type I cellulose secretion systems that are devoid of c-di-GMP-sensing BcsE orthologs resort to BcsR-homologous BcsP orthologs that could be essential for cellulose secretion

via the formation of synthase-interacting BcsPDQ cytoskeletal elements (experimentally demonstrated for *O. dioscraea* cellulose secretion (Sana et al. *Current Biology* 2024) (lines 489-493).

c) The different architectures of the Bcs complex represented in Gram-negative bacteria suggest there is more than one solution to one problem. What gives rise to these different architectures, or how are they differentiated (nutrient abundance in the niche? Accessibility of cellulose building blocks?)?

This is an excellent question that is the foundation for our future research. Different bacteria have evolved different Bcs architectures and likely resort to different cellulose packing and modifications for a structurally and functionally differentiated biofilm architecture. This could depend on nutrient abundance, eukaryotic hosts and co-evolved receptors or on expected microbial competition in the specific habitat.

d) Does activation of the synthase at lower c-di-GMP concentrations via the BcsE sensor yield similar catalytic efficiency (e.g. k_{cat}) as when micromolar concentrations of c-di-GMP are present? If yes, would it mean bacteria secrete cellulose at the same rate in different layers of the biofilm?

We are planning to address this in future works as we believe it needs to be correlated with subcellular and in biofilm macrocomplex expression and localization, cell morphology, BcsA and BcsERQ co-localization, and differentially disrupted BcsA vs. BcsE c-di-GMP binding affinities, both *in vitro* and *in situ*. The addition of excess Bcs(ER)Q have been shown to induce synthase catalytic activity in *in vitro* endpoint assays even in the absence of c-di-GMP and at levels comparable to that of c-di-GMP on itself (Acheson et al., NSMB 2021). As we show here, BcsEF act to stably retain BcsRQ to the macrocomplex. The above are consistent with the additional role of protein partner-mediated activation vs partner-independent c-di-GMP complexation and we have already demonstrated that the 'closed' BcsE conformation observed in both macrocomplex conformations is inducible by nanomolar c-di-GMP (Abidi et al. *Science Advances* 2021).

Reviewer #2 (Remarks to the Author):

Summary Comments to the authors:

The submitted manuscript by Anso et al. provides new details regarding the structural arrangement, stoichiometry and conformations of BcsF/E/R/Q subunits within the cellulose synthase macrocomplex. These subunits are also examined in light of the positioning of BcsA/B/G at the membrane interface and in many cases directly corroborate previously published indirect data and match presumed biological capacities. This work combines structural analysis by a number of means (cryo-EM, AlphaFold2 and crystal structure data manipulation) and is a remarkable feat, as it delineates for the first time important protein-protein interaction surfaces in the presence and absence of the c-di-GMP molecule that is typically viewed as an ON/OFF switch, but now should be viewed more broadly in the context of other overriding activation factors (including different complex conformations) across different c-di-GMP landscapes that contribute to biofilm formation.

We thank the reviewer for this great summary and the positive evaluation of our work. Again, we address the various specific comments below and thank the reviewer for her/his great attention to detail!

Main Comments:

The manuscript was well-written and provides a wealth of new information with regards to the cytosolic "vestibule" and its integration into the BcsAB complex. Importantly, the authors have recognized the presence and simultaneous submission of another manuscript that provides some overlapping structural data with regards to the cellulose synthase complex (i.e. reference 30 by Verma et al.). In many ways these manuscripts

corroborate each other and these have been noted well in the present manuscript, but there are some subtle differences (ex. BcsB interactions) that could use some clarification/reasoning/insight to address differences in experimental data or interpretation.

In all cases, these differences are understandable given the purification and analytical techniques used, but should still be noted for further work to clarify more fully rather than risk becoming dogma.

With regards to the BcsE conformations in BcsA bound/unbound c-di-GMP states, the authors focus on how this can allow polymer production even when c-di-GMP is absent from BcsA. This is still speculative and so the authors may want to be careful with how strongly they promote this. A more theoretical/suggestive viewpoint would be more pragmatic. Also, they fail to reference how this may be affecting BcsG activity. Given that BcsEFG are all part of the same operon and are in proximity to each other physically in the membrane (BcsG/F), it is feasible that these c-di-GMP states could also be affecting modification events.

Please comment on this possibility in the manuscript.

Indeed, we have refrained from commenting on BcsG activity as this is an *in vitro* study performed on cellulase-treated detergent-extracted samples and in the absence of phospholipids or UDP-glucose (i.e. BcsG or synthase substrates). It is important to note, however, that there are no immediate contacts between BcsG and its intraoperon partners in either conformational state (now mentioned explicitly in the text, lines 178-179) and that the pEtN-transferase is recruited by the N-terminal domain and C-terminus of the BcsA synthase, rather than its intraoperon partners. This is also confirmed by the Verma *et al.* study which demonstrates that the homodimeric BcsAB tandem from *R. sphaeroides* can on its own recruit a BcsG trimer just by the transplantation of *E. coli* BcsA^{NTD} on the synthase, i.e. in the absence of BcsE, BcsG and BcsRQ partners whatsoever, and in the absence of polymeric BcsB. It is also worth mentioning that *bcs* operons are highly mosaic: many Type II secretion systems feature all *E. coli* subunits in a single operon; other *bcs* operons have the regulatory *bcsR* and *bcsE* genes substituted by *bcsP* and *bcsD* genes encoding for cytoskeletal partners (with BcsP^{NTD} being homologous to BcsR; Sana *et al.* Current Biol 2024); yet in others the *bcsG* gene is substituted by a putative cellulose acetylation-determining *wss* cluster.

Overall the results on BcsG recruitment are consistent between the studies. It is important to note, however, that the Verma *et al.* study features at best secondary structure resolution data for the BcsA^{NTD}-BcsG interaction and all specific contacts are derived from low confidence AlphaFold models as opposed to the actual electron densities resolved at side-chain resolution in our study. The Verma *et al.* study also lacks resolved densities for the tail-anchors of BcsB copies 2-6 and completely lacks resolution of the BcsE, BcsF or BcsRQ partners even if they are present in the purified macrocomplex sample. However, we believe the two works are very complementary and not conflicting in any way: whereas the Verma *et al.* study provides further structure-function details on the periplasmic side of the macrocomplex with the quaternary structure of BcsZ and putative BcsC^{TPR}-crown interactions, our work provides high resolution insights into both the inner membrane-embedded and cytosolic regulators.

Overall we believe speculating about effects on BcsG activity other than its recruitment and stabilization within the macrocomplex primarily by BcsA^{NTD} and to lesser extent by the formation of the assembled macrocomplex per se is unwarranted. We have highlighted in the text the lack of contacts between BcsG and its intraoperon partners and have expanded the characterization of contacts involving BcsB. Thus all conclusions on interactions involving the BcsB tail anchors, BcsF, BcsE, BcsRQ and the cytosolic BcsA modules discussed here are based on well-resolved structural data uniquely presented in this study.

Finally, we have revised the manuscript to underscore that the effects of the higher affinity BcsE c-di-GMP sensor likely or possibly (and not certainly) provide an activation-by-proxy mechanism based on the structural

data presented here, on coevolutionary characteristics across BcsR/BcsP-dependent systems and on in vitro functional data regarding synthase activity and c-di-GMP-sensing affinities published previously by us and others.

Other Specific Minor Comments:

Note1: the use of line numbers would have helped in the review/edit process.

We have included line numbers in the revised manuscript.

Note2: Please check the figures and references for proper italics on bacterial names

We have double-checked the figures and references for the proper use of italics in bacterial names. We thank the reviewer for this attention to detail!

Note3: "Vestibule" - Not sure this is the right word for this as it represents a room or hallway next to an entrance. The BCSERQ complex is more like a doorman than a room, but I am not too fussed about this if the authors have a different interpretation and prefer the term.

As the BcsERQ complex forms around and stabilizes the PilZ module through multiple protein-protein interfaces without occluding the synthase active site or the c-di-GMP-binding PilZ domain linker we prefer to keep the term 'vestibule'.

1 – reword "host-associated microbes such as" to "host-associated microbes, such as"

We have edited the text as suggested (line 43).

2 - reword "transferase (GT) and an membrane transport" to "transferase (GT) and membrane transport"

Text edited to "a glycosyl transferase (GT) and a transmembrane export (TMD) domains" to indicate one of each functional modules (lines 67-68).

3 – pg2./2nd paragraph- you define the different Bcs systems. It was noticed later on that figure 6 has the acetylation machinery in the model. Perhaps here you could also address this modification with a sentence or two and which type it fits into?

We have added an additional supplementary figure on synthase fold conservation and cellulose secretion systems and mentioned the possibility of cellulose acetylation by an alginate acetylation-like Wss complex in the introduction (Supplementary Fig. 1 and lines 78-80)

4 – pg.3/2nd paragraph – In terms of stoichiometry it seems that most of the Bcs components are in pairs, but here it is noted that BcsG is a trimer. What is the relevance of this to the symmetry in the complex?

As we discuss in this letter and in the text/figures, it could be a compromise between available space (no more than three copies can be fitted) and enzyme efficiencies. Notably, BcsA is a monomer whereas BcsG is a trimer. BcsG uses (and also produces) lipids that are not normally enriched in the outer leaflet of the inner membrane so increase of BcsG copy number could contribute to the efficient and extensive modification of the processively extruded cellulose (lines 445-450).

5 – "Importantly, the higher-affinity BcsE sensor remains c-di-GMP-bound in both structures and retains a

compact conformation 15 to retain the regulatory BcsR2Q2 complex onto the synthase, likely preserving the latter's catalytically competent state." What about affects on BcsB or BcsG? Can you reword to be more encompassing?

BcsG does not contact BcsE or BcsF. We have however commented that a hexameric BcsB crown is stabilized in the presence of all subunits and wild-type macrocomplex assembly. The hexameric crown, although not fully resolved at the transmembrane level likely limits the positional heterogeneity of the BcsA^{NTD}-BcsG³ assembly, so the effects are indirect (lines 463-469)

However there is no significant difference in the crown conformation between the two states and as confirmed by the Verma et al. study BcsG₃ recruitment depends on BcsA^{NTD} and not on the organization of the BcsB crown. Therefore the focus here is on BcsE, whose conformation drastically depends on c-di-GMP and which is directly involved in retaining BcsRQ to the macrocomplex (as shown by the BcsF truncations here and by BcsE^{NTD} deletion in Zouhir et al. mBio 2020, for example)

6 – reword “has been shown the associate with” to “has been shown to associate with”

Corrected as suggested (line 122).

7 – pg.4/para3 - Figure 1 has individual coloring that should be referenced in brackets as you talk about the different components here. This will help the reader sift through the positioning of each player in the macrocomplex more readily.

We have preserved the color coding as much as possible throughout the manuscript and have added colors in brackets where appropriate.

8 – pg.5/para2 – “The first and second copies BcsB copies engage in contacts with the BcsA synthase”. Recent Verma et al. manuscript suggests 1 BcsB and BcsA interaction. Can you address this discrepancy?

The Verma et al study does not have any resolved BcsB anchors other than the co-catalytic anchor of the first BcsB copy that completes BcsA^{TMD} and which is present in all BcsAB tandems. The overall resolution of the Verma *et al.* study is overall at the secondary structure level for the inner membrane BcsAG3 complex and lacks any interpretable resolution for the cytosolic Bcs partners or other inner membrane components; specific contacts are inferred from low-confidence AlphaFold models. In contrast, our structures feature side-chain resolution for the BcsAG3 complex, as well as the cytosolic components, and has resolved densities for multiple BcsB anchors as well as for the BcsF dimers in the two captured states.

To avoid the conception of structural discrepancies rather than resolution limitations we have detailed the description of the BcsB-BcsA contacts to discriminate between the classical polymerase-copolymerase contacts between BcsA and BcsB1 and the more limited contacts with BcsB2 (lines 168-172).

9 – pg.7/para2 – “The cryo-EM structures presented here are consistent with the previously tripartite architecture of BcsE, comprising a degenerate trio of an NTPase*, REC* and GGDEF* domains (Fig. 1a)”. I would suggest moving this part of Figure 1 to Figure 3 to follow the order of the manuscript better and place it in better context.

We would also like to avoid referring to earlier figure panels, however we have preserved BcsE domain architecture in Figure 1 as its introduction is necessary for presenting the state-of-the-art and for identifying the modules and color coding of the separate domains in the macrocomplex assemblies.

10 - pg7 – reword “different orientations relative to the to the apical”

Reworded to remove the duplication (line 261).

11 – pg8/para2 - This importantly extends this work and further defines the different types of cellulose synthases. While the presence of a hydrophilic pocket is mentioned here and sequence conservation, a reason or importance of this is not noted for the type III. Is this likely to be a protein-protein interaction domain? What would the purpose purportedly be? This area is nicely compared for type 2 and type 1, but is lacking for type 3.

The presence vs. absence of a hydrophobic pocket is exactly what is important. Type III systems lack cytosolic partners of either the BcsERQ or BcsPDQ types, which appears to correlate with a hydrophilic surface on the synthase.

We have modified the text to underscore putative synthase-partner activator coevolution in enterobacteria, as well as the lack of cytosolic partners in Type III systems (lines 317-321).

12 – pg9/last para – reword “insights, we kept low micromolar concentration”

We have modified the sentence to indicate the dinucleotide concentration: “To gain mechanistic insights into the c-di-GMP-dependent regulation, we kept low micromolar concentration of the dinucleotide (2-4 μ M) throughout the purification procedure and prepared the cryogrids after a final fast concentration step to approximately 2:1 c-di-GMP:Bcs macrocomplex ratio” (lines 384-387).

13 – pg10/para2 – “X-shaped BcsF dimer is found shifted between the fourth and fifth BcsB protomer as opposed to the c-di-GMP-saturated complex”. Is there interaction between F and G or E and G in this conformation too that may influence PEtN modification of cellulose? Could it uncouple BcsG activity from BcsA possibly? Given the influence of BcsB and BcsG on BcsA activity, this alternate conformation may have an unidentified effect that is propagated at the membrane level and not through RQ? Expanding on more of this (especially if there is structural evidence/variation in support) and what is briefly mentioned in the next paragraph would be good.

As mentioned above, there are no direct interactions between BcsF and BcsG or between BcsE and BcsG and we feel discussing effects on BcsG catalytic activity in the context of an overall preserved BcsAB₆G₃ assembly between the two states presented here is unwarranted. Whereas we and others have shown over multiple studies that BcsA integrity depends on the majority of its macrocomplex partners, including BcsG, we have focused here on the specific, well-resolved differences in protein-protein interactions and their conserved features that are determined as key to cellulose secretion.

Overall, the c-di-GMP-free synthase features slightly better resolved densities for BcsG in the multicomponent assemblies. However, the locally refined BcsAG3 structures are closer in resolved features indicating that the difference in resolution likely comes from a more global conformational variability, rather than BcsG stabilization or activation per se. We have also indicated that the assembly of a full macrocomplex stabilizes a hexameric crown which can indirectly stabilize the position of the BcsG trimer in the assembly and is also dependent on the vestibule partners (lines 484-491).

However, it has been shown that the BcsB C-terminal anchor is sufficient for BcsA catalytic activity in vitro (Omadjela et al. PNAS 2013), we have further demonstrated that the periplasmic modules of BcsB are not absolutely required for the interaction and copurification of the synthase with the rest of the Bcs macrocomplex components (Krsteva et al., Nat Commun 2017) and the low-resolution cryo-EM structure of a chimeric synthase-BcsG complex in the Verma et al. study shows that a BcsA^{NTD} transplanted on the Rhodobacter

sphaeroides BcsAB tandem can recruit on its own a BcsG trimer, without any BcsB polymerization or cytosolic partners. So overall the coupling between cellulose synthase and pEtN-transferase activities seems to depend primarily on BcsA^{NTD}: BcsG interactions rather than on the intraoperon partners BcsEF or the periplasmic crown.

14 – pg11 – reword for better clarity “Conversely, older, stationary phase strata cascade-activate specific subsets”

We have reworded the sentence to: “Conversely, older, stationary-phase biofilm strata activate a cascade of c-di-GMP-metabolizing enzymes for gradual c-di-GMP increase, thus leading to non-dividing rounded cells embedded in a dense extracellular mesh of pEtN-cellulose and amyloid curli³” (lines 444-449)

15 – pg11/last para – “whose catalytic domain and enzymatic mechanism have been extensively characterized in vitro and in vivo^{13,21,23}.”. This paper (PMID: 34762795) I also tend to miss, but probably has the best in vitro characterization and should likely be mentioned here if you want to be comprehensive.

We thank the reviewer for highlighting this omission and have included the reference (line 465). Indeed work from the Weadge lab has been instrumental in the characterization of the periplasmic cellulose modification enzymes, including both BcsG and cellulose acetylation modules.

16 – pg11/12 – “The membrane sampling and significant conformational changes, which would be required for pEtN-transfer onto a polymer processively extruded through the periplasmic BcsB crown,”. It might also be pertinent to mention that the presence of three BcsG could simply be to try to match the catalytic rates of cellulose synthesis by BcsA with pEtN modification by BcsG in vivo, which are most likely not the same in a 1:1 scenario. Precise pEtN levels might be important for biofilm formation and maintenance and so complementary rates between BcsA and BcsG to achieve 50% modification would likewise be important to control and achieved by simply having 3 copies of BcsG.

We have modified the paragraph to underscore the trimer to monomer stoichiometry and the need of sufficient substrate mining in the context of extensive modifications of a processively synthesized polymer (lines 465-471). We would like however to avoid speculating about specific catalytic rates as these are not the subject of or compatible with the design of this study (see above).

Figure 1

- reword “have remained largely unresolved in the, whereas crystallographic snapshots”

Reworded

- 3rd panel of A - Unclear why we need the 90 degree rotation of BcsB here but nothing else? Maybe just show E in the splayed and closed conformations? Or keep this for the later figure on BcsE. Indeed it seems a bit out of place here in the manuscript.

The 90 degree rotation is to show that the crown is a hexamer. As this is not a real structure but a composite structure-thumbnailed model it is not straightforward to show the rotations for all elements. We have underscored that the top view serves to show two different views of the hexameric BcsB

The BcsE conformational changes are in contrast deduced from crystallographic snapshots and similarly are separate from the macrocomplex model where the protein's conformations have evaded characterization.

Overall, figure 1A shows the state-of-the-art, therefore we believe it requires all elements presented.

- at first glance the coloring of E and A makes this a bit confusing to follow. May think about simplifying this to match the operon more?

We have used conserved color-coding throughout the figures and our overall body of work. Some elements have multiple domains with specific functions, which requires the introduction of additional subsets of colors.

- Panel B – Better labelling of the following would help: 1. Where is the membrane? 2. E is a bit confusing across the panels 3. Label G and B in more than first panel.

We have added additional labels for the inner membrane, periplasm and cytosol. We have intentionally limited the labels where redundant or consistent across panels to make the figure less complex. BcsB is presented as in Figure 1a and we have added the color-coding for its separate domains to the figure legend.

- Panel C – More labelling here would help. Also, these conformational changes with respect to activation and activity should be expanded on a bit more. The authors have done a good job of explaining how ERQ locks into a position for activity, but how this works relative to these large macrocomplex conformational changes is still not entirely clear in the body of the text.

Similarly, we have intentionally limited the labels where redundant or consistent across panels to make the figure less complex and easier to spot overlaying vs. conformationally different parts. We have explained conformational changes and relevant interfaces in the higher-resolution subcomplex structures and have kept the orientation of the two subcomplexes as close as possible.

Finally, we have extended the accompanying supplementary figure (now Supplementary Fig. 2) to include overlays for the two conformations and close-ups for the drastically different vestibule organization.

Figure 2

- showing the location of the membrane in panel A would help for orientation

The membrane position is indicated.

Figure 3

- The ranges (colors) of the electrostatic and hydrophobic surfaces should be noted for the reader for clarity of interpretation.

The color keys are now included for both the Coulombic electrostatic potential and the lipophilicity.

- Location of membrane in panel A .

The membrane location is now indicated in the figure panel.

Figure 4

- Location of membrane in panels A and B

We have now indicated the inner membrane location in panels A and B.

We have also included a color key for the Coulombic electrostatic potential.

Finally, we caught an error in the labeling of the BcsR point mutations. We have corrected D15K to D21K in the figure panel to match the experimental mutation, text and figure legend.

Figure 5

- Location of membrane in panels A, B and C

We have indicated the membrane in A and B, however in C the complex is tilted to better visualize the conformation of BcsERQ so we have opted to omit the membrane label.

Figure 6

- "by the pEtN-transferase BcsG or by a multicomponent Wss cellulose acetylation complex". As noted earlier, this is introduced out of nowhere at the very end. I like that this is being a bit more comprehensive with possibilities, but you might want to add this into the introduction and discussion somewhere too.

Even just a sentence that it is a possibility that fits somehow into the Types of cellulose biosynthesis.

We have added the cellulose acetylation modification by a putative alginate acetylation-like Wss complex in the introduction and depicted it in Supplementary Figure S1.

Figure S1

- Is there any evidence of BcsC interaction with BcsB?

We do not have BcsC and BcsZ overexpressed with the macrocomplex and we have not observed stable copurification even in native context. The Verma et al. study suggests interaction between BcsC^{TPR1-2} and the last copy of BcsB of the crown based on densities appearing at very low visualization thresholds in the presence of the BcsC periplasmic domain.

Figure S4

- Indicate BcsE and BcsF coloring in the last two panels.

We have indicating the proteins as requested.

Reviewer #2 (Remarks to the Author):

I appreciate the time and effort that has gone into addressing all of the reviewer's questions from the first submission. I would also like to thank them for the time they took to provide detailed explanations of their standpoints in the rebuttal letter, especially when they didn't make the suggested changes. I feel that the manuscript is greatly improved and have only a few further minor comments listed below that I caught on a second read:

Line 100 – I understand the reasoning in the rebuttal letter to trim this back, however, I think that at least 2-3 more sentences are needed here to highlight the key results.

Line 211 – can “including on the C-terminal periplasmic module of BcsG” be put in brackets instead of dashes to help with reading through the sentence with so many dashes?

Line 369 – reword to “concentrations”

Line 743 – Reword “in as Coulombic”

Figures – need to delineate what IM means in all the figure legends. IM is large in figure 3 panel A. Also delineate what PE means in Figure 2 panel E.

Structural basis for synthase activation and cellulose modification in the *E. coli* Bcs secretion system

Itxaso ANSO^{1,2,3}, Samira ZOUHIR^{1,2,4}, Thibault G. SANA^{1,2} and Petya Violinova KRASTEVA^{1,2,*}

POINT-BY-POINT REPLY TO THE REVIEWERS

Reviewer #2

I appreciate the time and effort that has gone into addressing all of the reviewer's questions from the first submission. I would also like to thank them for the time they took to provide detailed explanations of their standpoints in the rebuttal letter, especially when they didn't make the suggested changes. I feel that the manuscript is greatly improved and have only a few further minor comments listed below that I caught on a second read:

We thank the reviewer for the suggestions and helping us to improve our manuscript. We address the comments below and thank the reviewer once again for her/his great attention to detail.

Line 100 – I understand the reasoning in the rebuttal letter to trim this back, however, I think that at least 2-3 more sentences are needed here to highlight the key results.

We have extended this paragraph in order to highlight the key results.

Line 211 – can “including on the C-terminal periplasmic module of BcsG” be put in brackets instead of dashes to help with reading through the sentence with so many dashes?

We have exchanged the dashes for the brackets as suggested.

Line 369 – reword to “concentrations”

Reworded as suggested (“concentration” to “concentrations”)

Line 743 – Reword “in as Coulombic”

We have deleted “in” from the sentence.

Figures – need to delineate what IM means in all the figure legends. IM is large in figure 3 panel A. Also delineate what PE means in Figure 2 panel E.

We have indicated the meanings of IM and PE as suggested.

Finally, we noticed the incomplete legend with regard to panel 4f and have now completed it to fully describe the presented data.

We have additionally corrected the description of the commercial secondary HRP-conjugated antibody (“donkey” to “rabbit”).

Finally, we have restructured the manuscript as editorially requested.